# GraGR: Gradient-Guided Graph Reasoner for Aligned and Interpretable GNNs

## Abstract

We propose the **GraGR** framework, which leverages gradients as reasoning signals to address two intertwined challenges in GNNs: (1) *node-level gradient inconsistency* across neighbors, and (2) *interpretability misalignment* between model training and explanations. GraGR's core modules detect and smooth conflicting per-node gradients via a *conflict loss* and Laplacian-based smoothing, and convert pairwise gradient inner-products into attention weights for message passing. We further introduce a *meta-gradient scaling* scheme (learnable task weights updated by hypergradients) to balance heterogeneous objectives when multiple tasks are present. Together, these components reduce local gradient misalignment and yield more stable, faithful explanations. We extend GraGR to **GraGR++** by adding **multi-pathway routing** (parallel routing pathways) and an **adaptive training scheduler** that gates gradient reasoning until base convergence. Importantly, we define *six gradient-derived node features* that quantitatively characterize a node's learning dynamics and offer interpretable insights. Experiments on benchmark datasets (Cora, Citeseer, PubMed, OGB-MolHIV) show that GraGR/-GraGR++ improve predictive performance and explanation coherence compared to baselines, while significantly reducing the proposed conflict energy. This work unifies optimization and interpretability in GNNs under a **gradient-as-reasoning paradigm**, making node-level dynamics both correctable and explainable.

## 1 Introduction

Graph neural networks (GNNs) excel at leveraging both structure and features for node- and graph-level prediction. Yet, their training often suffers from a subtle but critical failure mode: **node-level gradient inconsistency**. Why should two neighboring nodes, tightly linked in the same graph, push the parameters in conflicting directions? Even within a single objective, per-node gradients can differ in magnitude, direction, and temporal stability, producing oscillations and fragile updates (Liu et al., 2021). Such misalignment not only destabilizes optimization but also undermines interpretability: post-hoc explanations may highlight features that did not actually drive parameter updates. While prior work has linked **gradient misalignment** to *multi-task learning* interference (e.g., MGDA Désidéri (2012), PCGrad Yu et al. (2020), GradNorm Chen et al. (2018)), we argue that the more fundamental issue lies at the *node level*: how within-task and across-neighbor gradients interact. Our goal is a unified mechanism that both corrects these local conflicts and makes the correction signals themselves visible as interpretable, per-node reasoning. Recent advances in ante-hoc self-explanation (e.g., X-Node (Sengupta & Rekik, 2025)) reinforce the value of building structured, node-level contexts for faithful and human-readable explanations.

At the same time, **GNN interpretability** has become critical in high-stakes domains. Post-hoc explainers (e.g. GNNExplainer Ying et al. (2019), PGExplainer Luo et al. (2020)) identify subgraphs and features responsible for predictions, but these are often decoupled from the model's training dynamics. In practice, explanations may be *unstable* or *disconnected* from how the GNN was actually trained. For instance, two instances of similar classes may receive very different explanations even if their gradients were similar during training. This disconnect raises concerns about *interpretability misalignment*. Recent works attempt more structured explanations (e.g., XGNN Yuan et al. (2020), LOGICXGNN Geng et al. (2025), or gradient-adjusted GEAR Zhang et al. (2024)), but none jointly align training gradients with explanatory structure.

These observations motivate GraGR: a **unified gradient-as-reasoning** framework. We hypothesize that *leveraging gradient information directly in the GNN can both resolve training conflicts and produce aligned, interpretable representations*. Concretely, GraGR inserts gradient-guided modules into GNN layers. These modules detect and smooth out conflicting gradients (via a novel conflict loss and Laplacian smoothing), and use gradient inner-products to form an attention-like weighting over edges. Additionally, meta-learned scaling parameters adaptively re-weight tasks. Intuitively, GraGR treats gradients not just as optimization signals but as *latent explanations* that guide message passing. By coupling gradient alignment with reasoning, we aim to produce a GNN whose optimization trajectory is inherently interpretable.

## 2 PROBLEM STATEMENT AND HYPOTHESIS

Consider a GNN tasked with $T$ objectives (e.g., $T$ classification or regression losses) on a graph $\mathcal{G}$. Let $\{\mathcal{L}_i\}_{i=1}^T$ denote the task-specific losses. At a node $v$, define $g_i(v) = \nabla_{\mathbf{h}_v} \mathcal{L}_i$ as the gradient of task $i$ with respect to the node's representation $\mathbf{h}_v$. We identify two intertwined problems:

- **Task-level gradient conflict:** Gradients $\{g_i\}$ may point in divergent directions. Formally, tasks $i, j$ conflict if $g_i^\top g_j < 0$. In expectation over data, conflicting tasks hinder convergence to a joint optimum. We define a *gradient conflict energy* as

$$E_{\text{conf}} = \sum_{i<j} \max\left(0, -\frac{g_i^\top g_j}{\|g_i\| \, \|g_j\|}\right),$$

  which is positive when the cosine similarity is negative. Prior work Yu et al. (2020) Liu et al. (2021) shows that large $E_{\text{conf}}$ slows multi-task learning.

- **Node-level gradient inconsistency:** Within a single task, gradient magnitudes or directions may vary widely across neighboring nodes, especially in irregular graphs. This leads to unstable optimization, akin to overshooting or oscillation. It also causes explanation disconnect: an explainer may attribute importance to features that did not actually drive training on that instance. In particular, if gradients are noisy, post-hoc explanations (based on e.g. saliency) may not align with model reasoning.

We propose the hypothesis that **aligning and feeding back structured gradient information into the GNN can unify optimization and explanation**. Concretely, if we encourage gradients across nodes to be aligned, the network will both converge more smoothly and produce predictions with built-in, gradient-consistent explanations.

## 3 RELATED WORK

Recent graph neural network (GNN) studies identify a common problem: **conflicting learning signals** across nodes and scales. In multi-objective or multi-task settings (e.g. node-level vs. graph-level tasks or multiple self-supervised losses), gradients from different parts of the graph can point in opposing directions, destabilizing training. For example, Désidéri (2012) showed that self-supervised GNNs with diverse pretext tasks require multi-gradient descent (MGDA) to "minimize potential conflicts" among gradients. Similarly, Zhang et al. (2024) observe that explainers for GNNs must balance multiple objectives (fidelity, sparsity, connectivity, etc.), and that **"conflicts between the gradients"** of these objectives can lead to suboptimal solutions. In practice, conflicting node-wise signals can cause **instability** (oscillating or vanishing updates) during training.

**Multi-task learning in GNNs:** Training GNNs with multiple objectives often leads to gradient interference Liu et al. (2021). Techniques like the Multiple Gradient Descent Algorithm (MGDA) seek Pareto-optimal solutions Désidéri (2012), while PCGrad performs "gradient surgery" to project conflicting gradients onto compatible directions Yu et al. (2020). GradNorm adaptively balances task losses via gradient magnitudes Chen et al. (2018). In graph domains, recent works explore multi-task self-supervision: e.g., ParetoGNN Ju et al. (2023) uses MGDA to reconcile diverse pretext tasks. However, these methods treat gradients only as optimization signals and do not integrate interpretability.

**Gradient smoothing and topology-aware methods:** It is well-known that graph convolutions perform Laplacian smoothing of node representations Park & Kim (2024), which can both help and hurt training (oversmoothing). Some works modify graph topology (e.g. rewiring or adding edges) to alleviate bottlenecks. Our Laplacian Gradient Alignment component similarly diffuses gradient signals along the graph structure, smoothing out local conflicts. This is related in spirit to techniques that apply graph Laplacians for denoising or improving gradient flow, though GraGR uses the Laplacian to align multi-task gradients. Also, recent works apply meta-learning to multi-task weighting. For instance, MetaBalance adapts auxiliary loss weights by controlling gradients via a meta-objective He et al. (2022). GraGR's meta-gradient scaling is in this vein: we introduce learnable task scalars $\gamma_i$ updated by hyper-gradients, which effectively learn how to balance tasks during optimization.

**GNN interpretability:** Post-hoc explainers identify important subgraphs or features. GNNExplainer Ying et al. (2019) finds a compact subgraph maximizing mutual information with predictions. PGExplainer Luo et al. (2020) uses a parametric generator network to output explanatory masks for multiple instances. Model-level methods like XGNN Yuan et al. (2020) train a graph generator (via RL) to find prototypical patterns, while LOGICXGNN Geng et al. (2025) extracts human-readable logic rules from a GNN. These approaches, however, often ignore gradient dynamics during training. Recently, GEAR Zhang et al. (2024) introduced gradient adjustment for explainers: it identifies conflicts among fidelity, sparsity, and connectivity objectives and refines gradients to improve explanation optimization. GraGR is distinct in that it embeds gradient feedback into the GNN itself, aligning learning with reasoning. More recently, ante-hoc node-level explainable methods such as X-Node construct compact per-node contexts and decode them into natural-language rationales, highlighting the value of structured node representations for faithful explanations (Sengupta & Rekik, 2025).

## 4 METHODOLOGY

**Gradient-Guided Graph Reasoner (GraGR)** framework treats gradients as explicit reasoning signals for graph neural networks (GNNs) and subsequently **GraGR++** which comes with adaptive scheduling and mult-pathways optimization. GraGR augments standard GNN training with modules that *detect*, *align*, *re-weight*, and *schedule* gradient flows, yielding more stable optimization and interpretable reasoning. Formally, let $\mathcal{G} = (V, E)$ be a graph with $|V| = n$ nodes, $|E| = m$ edges, and let $h_v^{(l)} \in \mathbb{R}^d$ denote the embedding of node $v$ at layer $l$. The loss function is $L = \sum_i \mathcal{L}_i$, where each $\mathcal{L}_i$ may represent a task-specific objective. We denote the per-node gradient as $g_v = \nabla_{h_v} L \in \mathbb{R}^d$. During GNN training, GraGR monitors per-node gradients and enforces alignment across the graph and is built upon six key components as illustrated in Fig 1.

### 4.1 GRADIENT-AWARE CONFLICT DETECTION

The first step is to explicitly identify nodes whose gradients oppose the global learning direction. For node $v$, we define a *contextual gradient* $g_{\text{ctx}}(v)$, e.g. the average of neighbor gradients or the dominant principal gradient of the graph:

$$g_{\text{ctx}}(v) = \frac{1}{|\mathcal{N}(v)|} \sum_{u \in \mathcal{N}(v)} g_u.$$

A node $v$ is flagged as conflicting if *(See Appendix A.1 for examples)*

$$\|g_v\| > \tau_{\text{mag}} \quad \wedge \quad \cos(g_v, g_{\text{ctx}}(v)) < \tau_{\text{cos}}, \tag{1}$$

where $\tau_{\text{mag}}, \tau_{\text{cos}}$ are thresholds. Negative cosine similarity indicates destructive interference. We define a *conflict loss* to quantify disagreement:

$$L_{\text{conf}} = \sum_{(i,j) \in E} \max(0, -g_i^\top g_j). \tag{2}$$

Large $L_{\text{conf}}$ signals widespread gradient misalignment. For each conflicting node $v$, we project its gradient to remove the opposing component:

$$g_v' = g_v - \frac{g_v^\top g_{\text{ctx}}(v)}{\|g_{\text{ctx}}(v)\|^2} g_{\text{ctx}}(v). \tag{3}$$

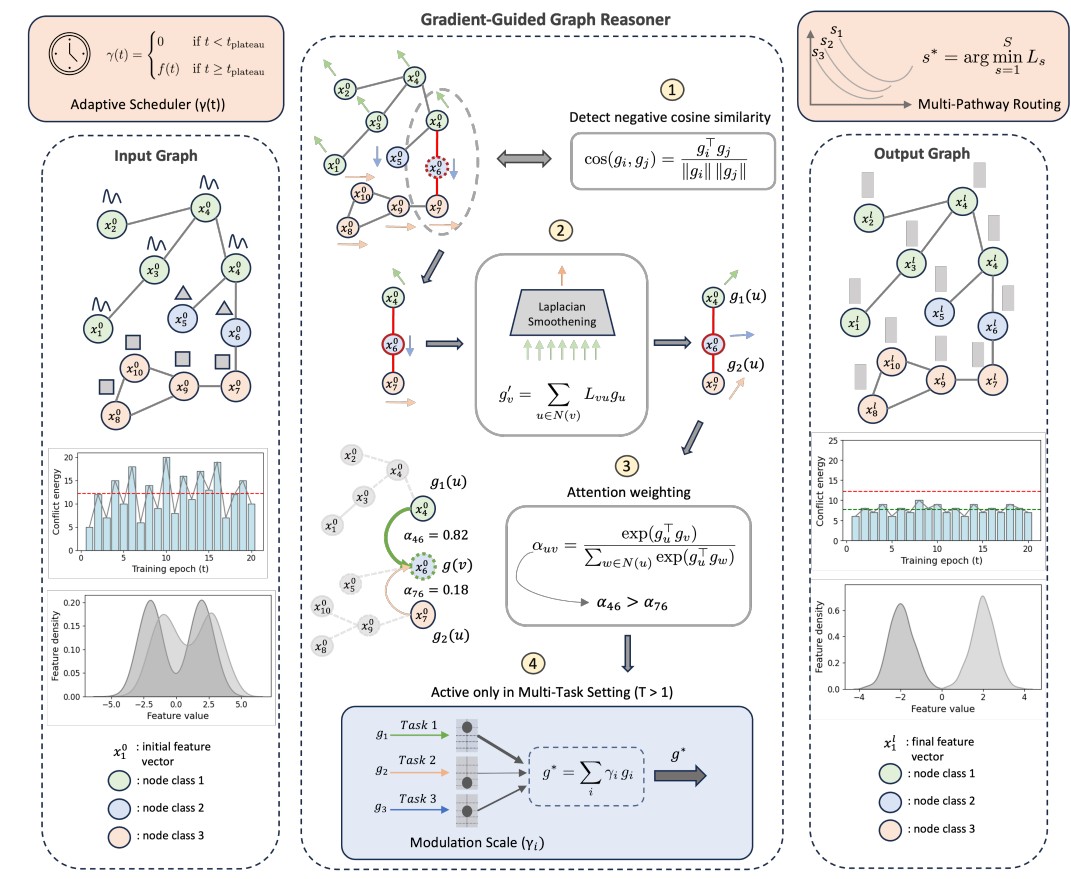

Figure 1: GraGR architecture and its extension to GraGR++ with adaptive scheduling and multi-pathway routing.

**Lemma 1 (Conflict Projection Validity)** *If $\cos(g_v, g_{ctx}) < 0$, then the projected gradient $g'_v$ satisfies $\cos(g'_v, g_{ctx}) \geq 0$. [Thus only conflict nodes are corrected, preserving non-conflict updates.] (See Appendix F)*

### 4.2 TOPOLOGY-INFORMED GRADIENT ALIGNMENT

To enforce global coherence, we smooth gradients across the graph topology. Let $L = D - A$ denote the combinatorial Laplacian of $\mathcal{G}$, with degree matrix $D$ and adjacency $A$ *(See Fig 5 and 7 in Appendix A.3)*. We seek adjusted gradients $\{g'_v\}$ by solving:

$$\min_{\{g'_v\}} \sum_v \|g'_v - g_v\|^2 + \lambda \sum_{(i,j) \in E} \|g'_i - g'_j\|^2. \tag{4}$$

The optimality condition yields:

$$(I + \lambda L)g' = g \quad \Rightarrow \quad g' = (I + \lambda L)^{-1}g,$$

which corresponds to low-pass filtering of gradients on the graph.

**Iterative Approximation.** We approximate the solution of $(I + \lambda L)g' = g$ via Jacobi iteration. Writing $(I + \lambda L) = D + R$, the update is

$$g^{(k+1)} = D^{-1}\big(g - R\, g^{(k)}\big),$$

which converges under standard spectral radius conditions *(see Appendix F)*.

**Lemma 2 (Gradient Smoothing Convergence)** *Under mild spectral conditions (e.g., $\rho(D^{-1}R) < 1$), the Jacobi iteration converges to the unique minimizer $g^\star = (I + \lambda L)^{-1}g$. (See Appendix F)*

This smoothing eliminates local conflicts while preserving global structure. Since GCNs already implicitly smooth features, applying smoothing to gradients aligns them analogously. We incorporate one step of Laplacian smoothing per layer in GraGR. The step costs $O(|E|)$ per gradient pass. For scalability on large graphs, sparse Jacobi or multigrid approximations can be used.

## 4.3 GRADIENT-BASED ATTENTION

We propose a novel *gradient-attention* mechanism that converts gradients into reasoning signals in the forward pass. At layer $l$, suppose nodes $u, v$ are connected. For each task $i$, let $g_i^{(l)}(u)$ and $g_i^{(l)}(v)$ denote the smoothed gradients with respect to the node representations. We define the attention weight as

$$\alpha_{uv}^{(l)} = \frac{\exp\left(\beta \sum_{i=1}^{T} g_i^{(l)}(u)^\top g_i^{(l)}(v)\right)}{\sum_{w \in \mathcal{N}(u)} \exp\left(\beta \sum_{i=1}^{T} g_i^{(l)}(u)^\top g_i^{(l)}(w)\right)}. \tag{5}$$

Here $\sum_{i=1}^{T} g_i^{(l)}(u)^\top g_i^{(l)}(v)$ aggregates agreement across all tasks' gradients. Intuitively, edges where node gradients align receive higher weight, emphasizing pathways consistent with shared learning signals and deemphasizing conflicting or noisy edges. *(See Fig 8 in Appendix A.3)*. The message-passing rule then becomes

$$H^{(l+1)} = \sigma\left(\sum_{v \in \mathcal{N}(u)} \alpha_{uv}^{(l)} W^{(l)} h_v^{(l)}\right). \tag{6}$$

**Theorem 1 (Attention Validity)** *For any finite graph $\mathcal{G}$, the coefficients $\{\alpha_{uv}^{(l)}\}$ form a valid probability distribution over neighbors. Moreover, under mild smoothness and alignment assumptions, reweighting by $\alpha_{uv}^{(l)}$ ensures a descent direction for the loss. (see Appendix F)*

## 4.4 META-GRADIENT MODULATION

To adaptively balance heterogeneous signals, we associate each task (or node group) with a meta-scalar $\gamma_i$. The overall training objective becomes

$$L_{\text{total}} = \sum_i \gamma_i \mathcal{L}_i + \lambda_{\text{conf}} L_{\text{conf}}. \tag{7}$$

Here, $\gamma_i$ serves as a learnable weight that scales the contribution of each loss term, while $L_{\text{conf}}$ penalizes misaligned gradients.

**Hypergradient Update.** Unlike fixed weights, $\gamma_i$ is updated by hypergradient descent. After each parameter update, we evaluate a validation objective $L_{\text{val}}$ and compute

$$\gamma_i \leftarrow \gamma_i - \eta \frac{\partial L_{\text{val}}}{\partial \gamma_i}. \tag{8}$$

This treats $\gamma$ as hyperparameters in a bi-level optimization: the inner loop updates model weights, while the outer loop updates $\gamma$ to improve validation performance. In effect, $\gamma_i$ learns to suppress losses that generate conflicting gradients and amplify those that yield reliable progress.

**Alternative Interpretation.** Instead of $L_{\text{val}}$, one may use the conflict objective $L_{\text{conf}}$ for hypergradient updates:

$$\gamma_i \leftarrow \gamma_i - \eta \frac{\partial L_{\text{conf}}}{\partial \gamma_i}.$$

This perspective emphasizes $\gamma$ as local gradient modulators, directly tuned to reduce variance between node updates. Conceptually, this resembles GradNorm Chen et al. (2018) and related meta-balancing schemes, but here the modulation arises from explicit hyper-optimization.

**Theorem 2 (Meta-Scaling Convergence)** *Under standard smoothness and stability assumptions (see Appendix F), the hypergradient update on $\gamma$ converges to a stationary point of the validation objective, corresponding to a Pareto-stationary task balance.*

This meta-scaling allows GraGR to dynamically reweight signals during training. By amplifying informative objectives and suppressing harmful ones, the system learns a Pareto-stable tradeoff across tasks. Empirically, we will show that this modulation reduces gradient conflict energy $E_{\text{conf}}$ and improves multi-objective convergence.

## 5 GRAGR++: ENHANCING ROBUSTNESS

### 5.1 MULTI-PATHWAY ROUTING FOR CONDITIONAL REASONING

Training graph neural networks (GNNs) with reasoning modules often suffers from two challenges: (i) sensitivity to random initialization, leading to unstable optimization, and (ii) the difficulty of dynamically activating appropriate reasoning mechanisms across heterogeneous graph regions. We propose the *Multiple Pathways* framework, which integrates a **two-stage training strategy** with a **multipathway reasoning architecture**. This design ensures both stability (by selecting favorable training trajectories) and interpretability (by enabling conditional reasoning).

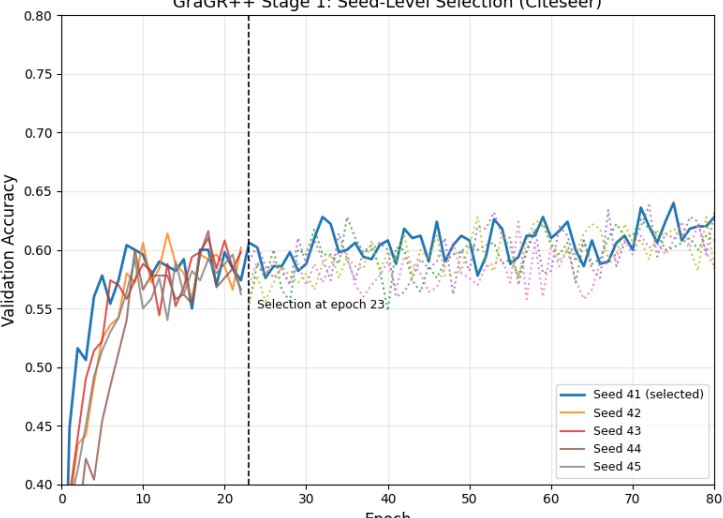

Figure 2: **Seed-Level Selection.** Validation accuracy trajectories across multiple random seeds on the Citeseer dataset. At the plateau boundary ($T_1$, dashed vertical line), the seed with the highest validation accuracy ($s^*$) is selected for subsequent training.

**Stage 1: Pathway Selection Across Random Seeds.** At initialization, different random seeds $s \in \{1, 2, \ldots, S\}$ generate diverse optimization trajectories. For each seed $s$ and epoch $t \in \{1, \ldots, T_1\}$, we record the validation loss:

$$\ell_s(t) \in \mathbb{R}. \tag{9}$$

Each trajectory is summarized by its final validation loss at $T_1$:

$$L_s = \ell_s(T_1). \tag{10}$$

We then select the best-performing trajectory:

$$s^* = \arg \min_{s \in \{1,\ldots,S\}} L_s. \tag{11}$$

This ensures that subsequent reasoning is applied only to a trajectory with sufficient convergence signal, reducing the risk of over-correcting noisy representations (see Fig. 2 for seed-level selection). In essence, *reasoning must be earned, not assumed*.

**Stage 2: Multipathway Reasoning Within the Model.** Once a stable seed trajectory $s^*$ is chosen, we extend the GNN with a specialized reasoning pathway for conflict-resolution, following the GraGR++ method. While other reasoning pathways (e.g., fidelity-preservation, connectivity-enhancement) are conceptually possible, they are not implemented in this work. Let $P$ denote the

set of logical pathways (e.g., conflict-resolution, fidelity-preservation, connectivity-enhancement). At layer $l$, each pathway $p \in P$ has parameters $W_p^{(l)}$ and computes an output:

$$H_p^{(l+1)} = f_p\left(H^{(l)}, A_p; W_p^{(l)}\right), \tag{12}$$

where $A_p$ is an adjacency mask or feature filter specific to pathway $p$. The representation at layer $l+1$ is updated via a gating mechanism for the conflict-resolution pathway $p_{\text{conf}}$:

$$H^{(l+1)} = (1 - \beta^{(l)})H_{\text{base}}^{(l+1)} + \beta^{(l)}H_{\text{conf}}^{(l+1)}, \tag{13}$$

where $H_{\text{base}}^{(l+1)}$ is the standard GNN update and $H_{\text{conf}}^{(l+1)}$ is the GraGR++ conflict-resolution update. The gating weight $\beta^{(l)}$ is dynamically increased when the conflict energy $E_{\text{conf}}$ exceeds a threshold. During training of the selected trajectory $s^*$, we track $E_{\text{conf}}$ over epochs. The GraGR++ pathway is activated only at epochs where $E_{\text{conf}}$ exceeds a predefined threshold, allowing *targeted conflict resolution* without affecting stable updates.

**Lemma 3 (Path Selection Criterion)** *If the update direction $d_{p^*}$ chosen by minimizing conflict (or maximizing agreement) satisfies $\nabla L_{total}^\top d_{p^*} < 0$, then $d_{p^*}$ is a descent direction for $L_{total}$. (See Appendix F)*

## 5.2 Adaptive Scheduling for Efficient Training

Running gradient reasoning at every epoch can be unnecessary, and even harmful, in the early stages of training when the base GNN is still learning low-level representations. We therefore introduce a scheduler $\gamma(t)$ that *activates reasoning only once the base model has plateaued*, ensuring stable embeddings before applying more complex corrections.

**Gate Definition.** Let $\mathcal{L}_{\text{base}}(t)$ denote the loss of the base GNN at epoch $t$, and define the one-step improvement

$$\Delta\mathcal{L}_{\text{base}}(t) = \mathcal{L}_{\text{base}}(t-1) - \mathcal{L}_{\text{base}}(t).$$

The reasoning gate is then

$$\gamma(t) = \begin{cases} 1, & \Delta\mathcal{L}_{\text{base}}(t) \leq \eta_{\text{thresh}} \ \wedge \ t \geq t_{\min}, \\ 0, & \text{otherwise}, \end{cases} \tag{14}$$

where $\eta_{\text{thresh}}$ is a small threshold (detecting plateau) and $t_{\min}$ is a warm-up period to allow the base GNN to stabilise.

**Training with Gating.** When $\gamma(t) = 1$, conflict detection, alignment, and gradient-based reasoning are applied; otherwise, training proceeds with the base GNN alone:

$$\mathcal{L}_{\text{corr}}(t) = \begin{cases} \mathcal{L}\big(h^{\text{GraGR}}(t)\big), & \gamma(t) = 1, \\ \mathcal{L}_{\text{base}}(t), & \gamma(t) = 0. \end{cases}$$

This prevents reasoning parameters from being updated prematurely, and saves computation whenever $\gamma(t) = 0$. *(See Appendix D for full algorithm)*

## 6 Results and Benchmarks

Across the six node-classification benchmarks *(Refer Table 4) in Appendix)*, GraGR and GraGR++ consistently boost baseline GNNs, with most improvements evident in GCN and SAGE models (Table 1). Gains are particularly pronounced on challenging datasets such as Cornell, Texas, and Wisconsin, where vanilla baselines converge to much lower accuracies, while GraGR++ models achieve substantially higher validation and test scores. GIN shows less stability, with GraGR occasionally underperforming, suggesting sensitivity to architecture-specific dynamics. The validation accuracy curves in Figure 4 *(See Appendix A.2)* further illustrate this trend: GraGR and GraGR++ converge faster and to higher plateaus than baselines, especially in high-variance datasets like Cornell and CiteSeer. Notably, GraGR++ occasionally underperforms relative to GraGR, which may be attributed to its added complexity and higher sensitivity to noisier datasets. *(see Appendix E for Ablation Studies and Appendix G for Computational Analysis)*

Table 1: Results of baseline GNNs vs GraGR vs GraGR++ models across 6 datasets. Best Test/Val values within each model family are highlighted in green.

| Model | CiteSeer | | | | Cora | | | |
|---|---|---|---|---|---|---|---|---|
| | Test | Val | F1 | AUC | Test | Val | F1 | AUC |
| GCN | 54.02 | 60.27 | 51.03 | 86.01 | 72.95 | 71.68 | 72.94 | 92.97 |
| GCN + GraGR | 65.01 | 67.43 | 61.12 | 86.57 | **79.82** | **78.82** | 78.22 | 94.98 |
| GCN + GraGR++ | **67.21** | **67.83** | 63.23 | 88.22 | 77.11 | 78.01 | 75.92 | 94.56 |
| GAT | 65.72 | 67.63 | 60.83 | 85.87 | 77.55 | 76.63 | 77.01 | 95.83 |
| GAT + GraGR | 63.74 | 65.21 | 59.85 | 83.61 | **78.82** | **77.63** | 77.45 | 95.34 |
| GAT + GraGR++ | **67.62** | **67.84** | 63.32 | 84.91 | 67.41 | 68.41 | 68.15 | 92.91 |
| GIN | **50.01** | **50.21** | 47.85 | 77.94 | **66.25** | **63.41** | 64.13 | 89.32 |
| GIN + GraGR | 25.32 | 23.62 | 9.94 | 51.71 | 42.21 | 39.61 | 37.42 | 83.96 |
| GIN + GraGR++ | 23.23 | 26.25 | 16.45 | 56.87 | 52.03 | 52.22 | 54.91 | 85.13 |
| SAGE | 60.71 | 64.02 | 57.45 | 84.31 | 78.55 | 76.81 | 76.91 | 94.63 |
| SAGE + GraGR | 67.15 | 68.01 | 61.34 | 86.59 | 79.26 | 76.67 | 79.19 | 94.23 |
| SAGE + GraGR++ | **67.44** | **68.92** | 62.93 | 82.83 | **79.64** | **79.23** | 79.12 | 94.82 |

| Model | Cornell | | | | PubMed | | | |
|---|---|---|---|---|---|---|---|---|
| | Test | Val | F1 | AUC | Test | Val | F1 | AUC |
| GCN | 35.11 | 52.52 | 21.65 | 62.15 | 68.15 | 66.02 | 66.51 | 87.12 |
| GCN + GraGR | 29.72 | 51.93 | 9.42 | 50.12 | **77.22** | **79.62** | 76.41 | 90.26 |
| GCN + GraGR++ | **51.44** | **55.91** | 24.92 | 51.65 | 76.91 | 79.02 | 76.32 | 88.94 |
| GAT | 37.82 | 54.23 | 13.75 | 56.52 | 76.73 | 78.62 | 75.23 | 87.26 |
| GAT + GraGR | **43.21** | **55.91** | 17.42 | 58.43 | **77.05** | **81.42** | 76.21 | 89.62 |
| GAT + GraGR++ | 40.51 | 55.63 | 14.53 | 49.02 | 76.62 | 80.21 | 75.41 | 89.13 |
| GIN | 45.93 | 57.34 | 36.55 | 67.41 | 49.13 | 49.83 | 42.63 | 67.25 |
| GIN + GraGR | 40.53 | 52.55 | 11.41 | 36.01 | 63.42 | 66.02 | 62.32 | 80.04 |
| GIN + GraGR++ | **51.41** | **58.91** | 24.65 | 46.34 | **68.53** | **67.41** | 66.62 | 81.05 |
| SAGE | 51.42 | 55.91 | 24.74 | 48.92 | 73.42 | 73.82 | 70.91 | 85.02 |
| SAGE + GraGR | 54.11 | 67.81 | 38.32 | 67.92 | 74.31 | 77.62 | 72.93 | 85.42 |
| SAGE + GraGR++ | **75.72** | **81.42** | 66.43 | 81.61 | **74.52** | **78.22** | 73.63 | 85.83 |

| Model | Texas | | | | Wisconsin | | | |
|---|---|---|---|---|---|---|---|---|
| | Test | Val | F1 | AUC | Test | Val | F1 | AUC |
| GCN | 48.62 | 61.01 | 22.31 | 62.04 | 39.22 | 50.03 | 21.31 | 56.61 |
| GCN + GraGR | **64.92** | 59.32 | 19.72 | 63.94 | 52.94 | 55.03 | 29.94 | 64.32 |
| GCN + GraGR++ | 59.51 | **62.72** | 19.65 | 61.04 | **53.32** | **57.51** | 31.91 | 61.22 |
| GAT | 59.51 | 54.22 | 15.22 | 62.95 | 52.94 | 57.51 | 13.85 | 68.71 |
| GAT + GraGR | 64.92 | 52.51 | 20.01 | 63.84 | 49.02 | 60.01 | 19.12 | 63.12 |
| GAT + GraGR++ | **67.63** | **59.31** | 29.92 | 65.92 | **54.91** | **60.31** | 19.41 | 63.81 |
| GIN | 51.42 | 54.42 | 17.92 | 62.91 | 45.13 | 52.53 | 17.92 | 51.82 |
| GIN + GraGR | 64.92 | 52.53 | 19.72 | 61.91 | 51.01 | 56.32 | 27.12 | 60.91 |
| GIN + GraGR++ | **65.81** | **57.52** | 21.74 | 65.94 | **54.03** | **57.23** | 29.62 | 65.21 |
| SAGE | 73.01 | 77.92 | 66.21 | 72.01 | 45.12 | 55.02 | 14.75 | 42.34 |
| SAGE + GraGR | **75.71** | **79.23** | 48.54 | 74.12 | 64.71 | 76.32 | 42.82 | 82.15 |
| SAGE + GraGR++ | 56.83 | 66.12 | 27.74 | 68.01 | **65.83** | **77.53** | 44.52 | 84.92 |

We evaluated GraGR's Multi-task learning on three classification datasets: OGB-MolHIV, PRO-TEINS, and MUTAG, each configured with five tasks. Table 2 summarises the results. On **MolHIV**, GraGR achieves the highest accuracy (0.626), clearly outperforming all baselines, while CAGrad provides the second-best performance (0.545). On **PROTEINS**, PCGrad yields the strongest accu-

Table 2: Multi-task classification results on OGB-MolHIV, TUDataset PROTEINS, and TUDataset MUTAG. Best results are highlighted in dark green; second-best in light green.

| Method | Final Loss | Accuracy |
|---|---|---|
| Vanilla Average | 4.535 | 0.350 |
| CAGrad | 4.538 | 0.545 |
| GradNorm | 4.563 | 0.429 |
| PCGrad | 4.538 | 0.361 |
| GraGR | 4.575 | 0.626 |
| GraGR++ | 4.620 | 0.299 |

**OGB-MolHIV** (5 tasks)

| Method | Final Loss | Accuracy |
|---|---|---|
| Vanilla Average | 4.427 | 0.332 |
| CAGrad | 4.426 | 0.372 |
| GradNorm | 4.460 | 0.295 |
| PCGrad | 4.430 | 0.507 |
| GraGR | 4.436 | 0.333 |
| GraGR++ | 4.436 | 0.374 |

**TUDataset PROTEINS** (5 tasks)

| Method | Final Loss | Accuracy |
|---|---|---|
| Vanilla Average | 4.885 | 0.411 |
| CAGrad | 4.888 | 0.473 |
| GradNorm | 4.903 | 0.355 |
| PCGrad | 4.874 | 0.225 |
| GraGR | 4.906 | 0.480 |
| GraGR++ | 4.865 | 0.571 |

**TUDataset MUTAG** (5 tasks)

racy (0.507), with GraGR++ ranking closely as the second-best (0.374). On the smaller **MUTAG** dataset, GraGR++ stands out with the highest accuracy (0.571), followed by GraGR (0.480). These results highlight that GraGR variants are highly competitive across tasks, consistently securing top or second-best positions, while baseline methods such as MGDA and GradNorm often lag behind.

## 7 FROM GRADIENT DYNAMICS TO INTERPRETABLE FEATURES

While GraGR primarily addresses gradient conflict in GNN optimization, its deeper contribution lies in rendering gradients into interpretable signals. If gradients encode how each node contributes to learning, then constraining and decomposing them provides not only stability but also human-understandable insights into the model's behaviour. Prior work in vision has shown that gradients can reveal salient input features Xuanyuan et al. (2022), and initial GNN explainers have explored gradient-based attribution Simonyan et al. (2014). GraGR enables an *interpretability-aware training regime*: the same gradients used for optimization can be re-purposed to explain what the model learns, when, and where in the graph. This is aligned with recent calls for *explanation-aware training* Sengupta & Rekik (2025), but extended here to gradient dynamics. *(Refer Appendix B)*

### 7.1 INTERPRETABLE GRADIENT CONTEXTS AND LLM DECODING

To produce per-node, human-readable explanations grounded in training dynamics, we summarize each node $v$ using six gradient-derived features *(see Appendix B for full definitions)*. We collect these into a compact context vector

$$c_v = \big[\psi_{\text{conflict}}(v),\ \psi_{\text{stability}}(v),\ \psi_{\text{influence}}(v),\ \psi_{\text{confidence}}(v),\ \psi_{\text{role}}(v),\ \psi_{\text{receptiveness}}(v)\big] \in \mathbb{R}^6. \quad (15)$$

This context is mapped through a lightweight Reasoner network (MLP) to obtain an explanation embedding

$$e_v = \text{Reasoner}_\phi(c_v) = W_2\,\sigma(W_1 c_v + b_1) + b_2, \quad (16)$$

which serves as both (i) an auxiliary signal for the GNN classifier and (ii) structured input for a large language model (LLM). Given a prediction $\hat{y}_v$, we query the LLM with a formatted prompt:

$$T_v = \text{LLM}\big(\text{prompt}(c_v, \hat{y}_v)\big), \quad (17)$$

yielding a natural-language explanation $T_v$ for node $v$.

---

**LLM Prompt for Gradient Context**

Node ID: $v$; Context vector: $c_v$
Prediction: $\hat{y}_v$; True label: $y_v$ (Optional)
Task: Generate a short natural-language explanation of the prediction based on the context vector. If the context indicates low reliability, suggest a possible corrective action.

---

The outputs of this prompt are concise, node-level explanations that expose how each node's gradient dynamics shaped its prediction. Illustrative examples, including both conflict-prone (Node 42) and stable node (Node 17), are provided in *Appendix B.1 (Interpretability Explanations)*.

## 8 CONCLUSION

We introduced **GraGR**, a *gradient-as-reasoning* framework that unifies optimization stability and interpretability in GNNs. By aligning gradients, GraGR/GraGR++ not only reduce conflict energy but also yield **ante-hoc, node-level explanations** via interpretable gradient contexts. Experiments across benchmarks show consistent gains in both performance and explanation coherence. This work moves toward *trustworthy, explanation-aware graph learning*, where gradients act as both optimization signals and human-readable reasoning.

Looking ahead, we ask: can gradient contexts be extended to capture *causal* influences in dynamic or heterogeneous graphs? Exploring such directions Mukherjee et al. (2025) may bring us closer to models that not only predict reliably but also **reason in forms humans can follow**

## REPRODUCIBILITY AND DEMOS

Our implementation of **GraGR/GraGR++**, along with scripts to reproduce all experiments, is publicly available at: `https://anonymous.4open.science/r/GraGR-30D7`. Comprehensive experimental settings, including dataset details, preprocessing, evaluation models, hyperparameter values, fine-tuning techniques, and hardware/software configurations, are provided in the Appendix section in Table 5.

We also release visual demonstrations of GraGR's behavior, such as gradient transformation and conflict healing animations. (Videos must be downloaded to view, as they do not play directly in-browser.)

- `https://anonymous.4open.science/r/GraGR-30D7/visualizations/gragr_transformation1.mp4`
- `https://anonymous.4open.science/r/GraGR-30D7/visualizations/gragr_healing_graph.mp4`

Further time and space complexities are illustrated in **Appendix C**, while computational analysis included in **Appendix G**.

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

# A ADDITIONAL PROOFS AND QUALITATIVE EXAMPLES

## A.1 EMBEDDING PLOTS SHOWING DETECTION OF CONFLICT NODES

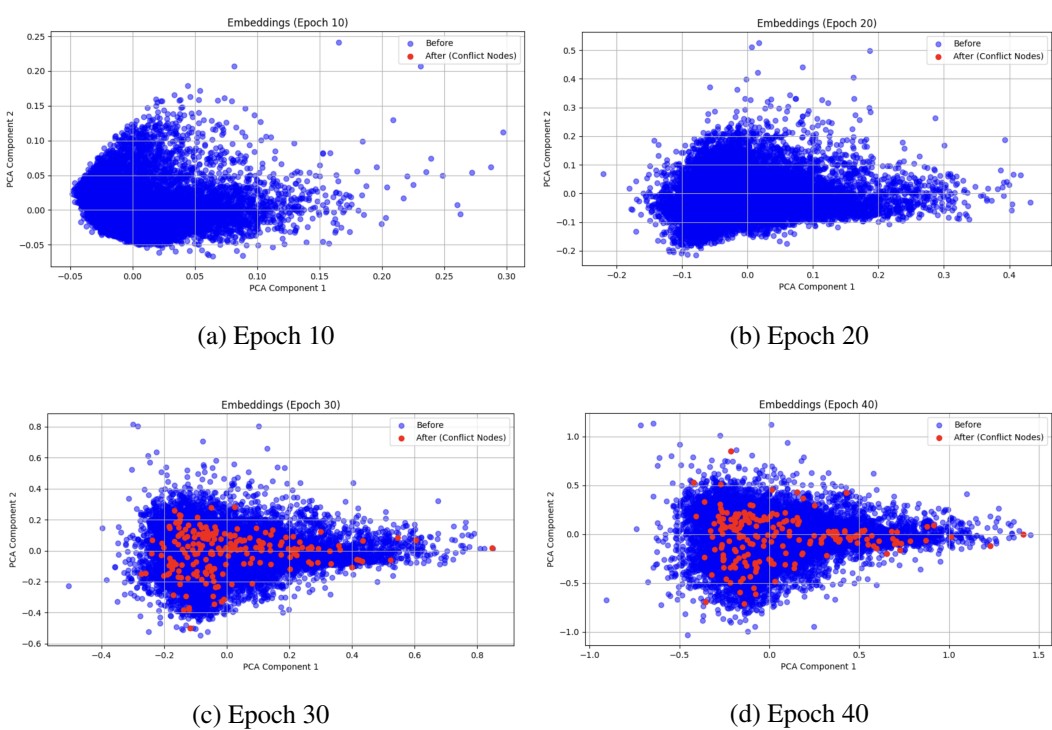

(a) Epoch 10      (b) Epoch 20

(c) Epoch 30      (d) Epoch 40

Figure 3: Embedding space of PubMed dataset showing detection of conflict nodes after GraGR++ getting activated at Epoch 30 (due to the adaptive scheduler concept).

## A.2 VALIDATION ACCURACY PLOTS

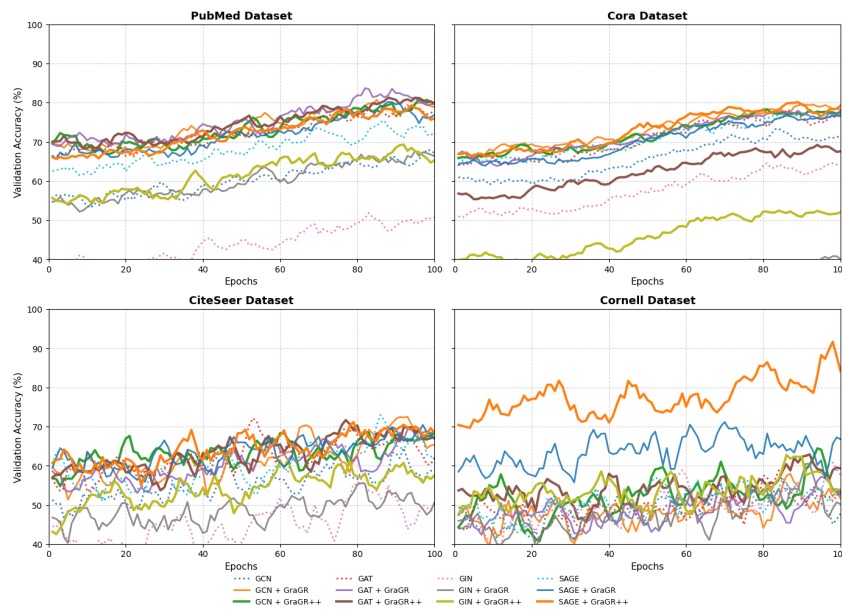

Figure 4: Validation accuracy over 100 epochs for 4 datasets (PubMed, Cora, CiteSeer, Cornell). GraGR and GraGR++ consistently achieve higher and faster convergence compared to baselines.

A.3   GRAGR COMPONENTS EFFECT ANALYSIS - CORA DATASET

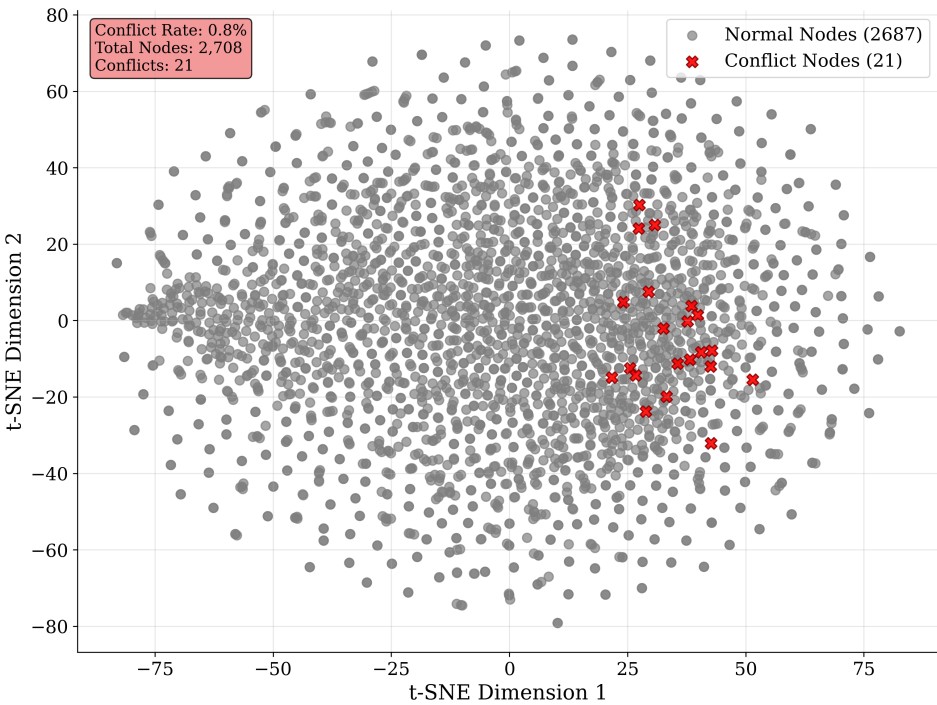

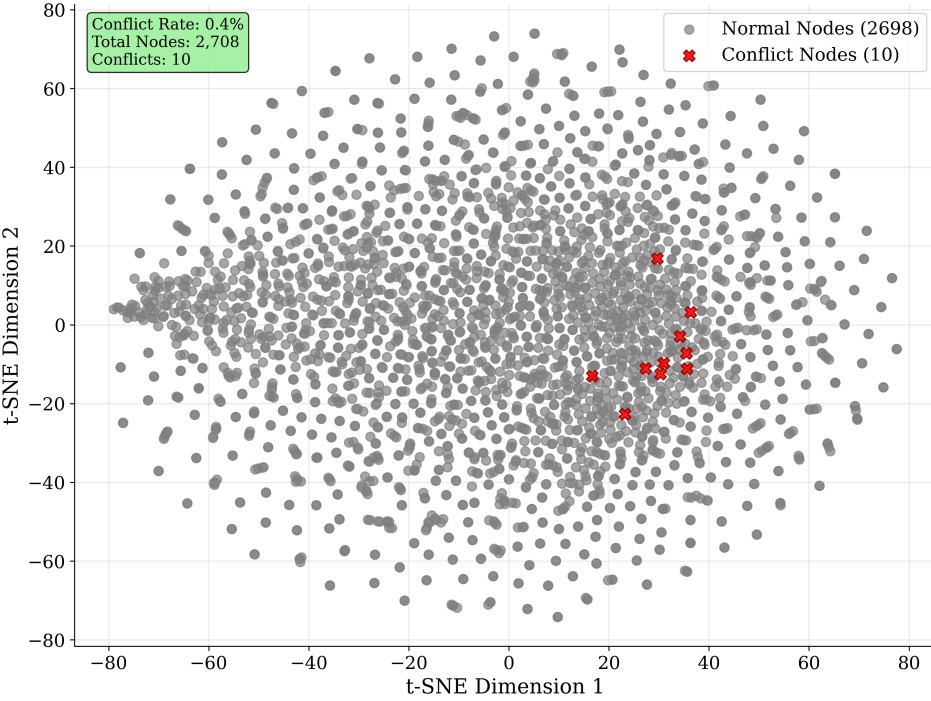

Figure 5: Node embeddings on the Cora dataset after Laplacian smoothing in a **Single Epoch**. Top: embeddings before conflict reduction, with conflicted nodes highlighted in red. Bottom: embeddings after conflict reduction using GraGR.

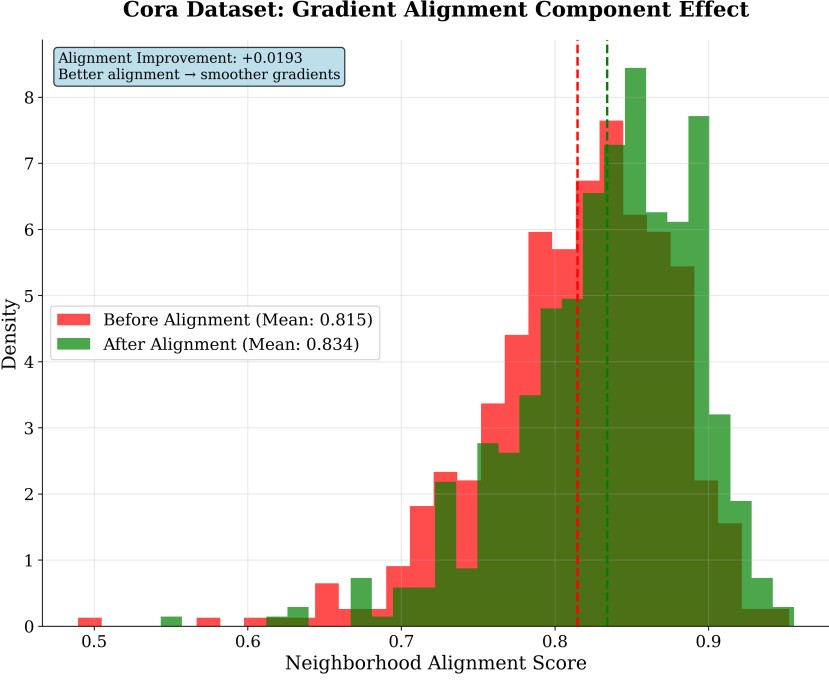

Figure 6: **Gradient Alignment**. Distribution of neighborhood alignment scores before and after applying GraGR. This highlights how gradient alignment improves local consistency in node representations.

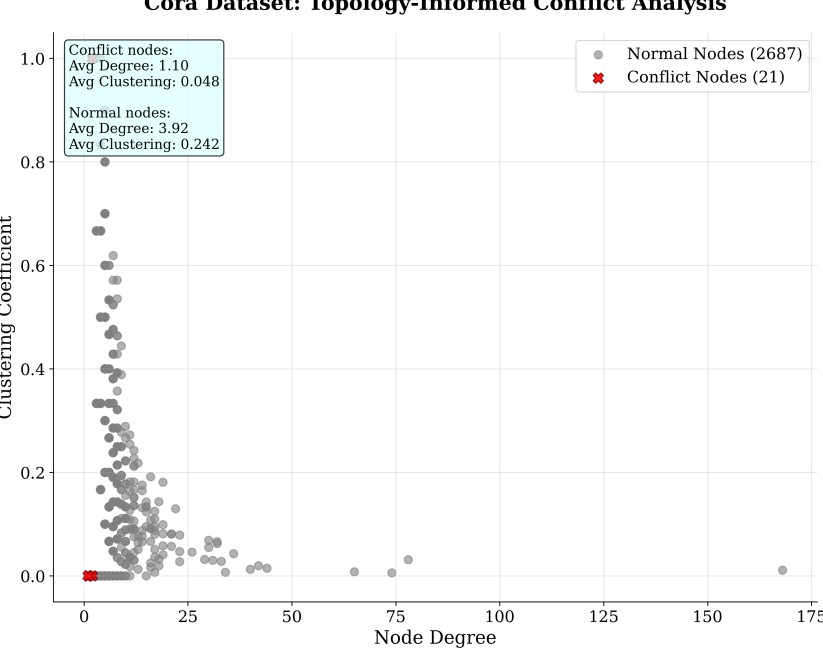

Figure 7: **Topology-Informed Processing**. Scatter plot of node degree versus clustering coefficient, illustrating how conflict nodes relate to graph topology. This demonstrates GraGR's topology-aware conflict detection.

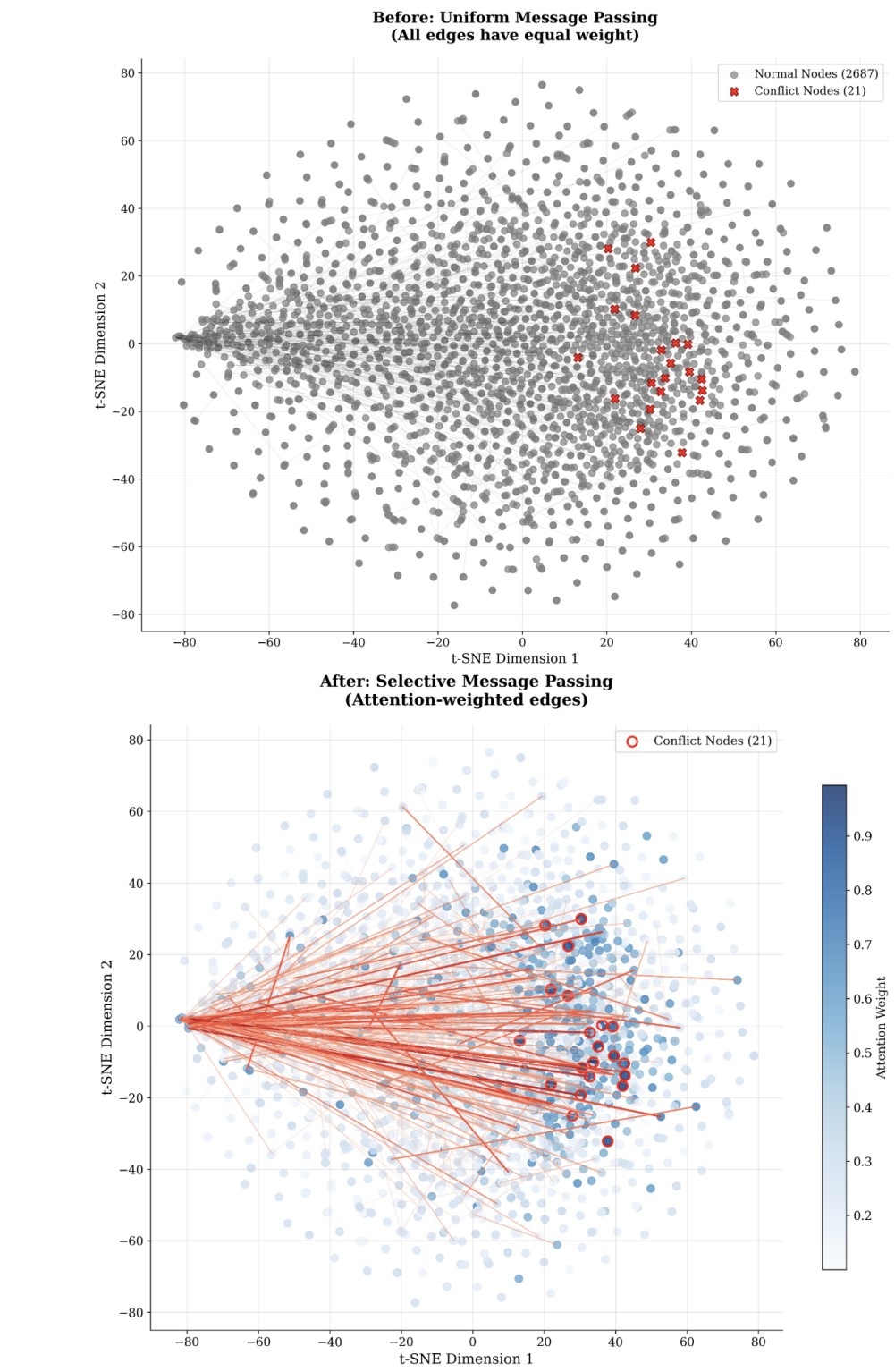

Figure 8: **Gradient-Based Attention** component of GraGR. The upper image shows uniform message passing where all edges have equal weight (grey) to the conflict node. The lower image illustrates how gradient-based attention reweights the edges, thereby altering the message aggregation to the conflict node.

## B   INTERPRETABILITY

To understand how and where GNNs learn, we propose mapping each node's raw gradient into a six-dimensional feature space: (1) conflict intensity, (2) trajectory stability, (3) multi-hop influence, (4) confidence–gradient alignment, (5) topological learning role, and (6) correction receptiveness. These features can be monitored *ante-hoc* (during training) or *post-hoc* (after training), enabling continuous interpretability of the learning process.

**a.  Gradient Conflict Intensity ($\psi_{\text{conflict}}$)**   This feature quantifies local disagreement between a node's learning signal and its neighbors. Formally:

$$\psi_{\text{conflict}}(i) = \mu_g \, \|\mathbf{g}_i\| \, (1 - \cos(\mathbf{g}_i, \bar{\mathbf{g}}_i)) \tag{18}$$

Here $|g_i|$ is the $\ell_2$ norm of the gradient at node $i$, $\bar{g}_i = \frac{1}{|N(i)|} \sum_{j \in N(i)} g_j$ is the average neighbor gradient, and $\mu_g = \frac{1}{N} \sum_{j=1}^{N} |g_j|$ is the global mean magnitude. The function $\cos(a, b) = a^\top b / (|a||b|)$ denotes cosine similarity.

**b.  Learning Trajectory Stability ($\psi_{\text{stability}}$)**   This feature measures coherence of gradient directions among a node's neighbors, reflecting neighborhood harmony. We define:

$$\psi_{\text{stability}}(i) = \frac{1}{\binom{|\mathcal{N}(i)|}{2}} \sum_{\substack{j,k \in \mathcal{N}(i) \\ j < k}} \cos(\hat{\mathbf{g}}_j, \hat{\mathbf{g}}_k) \tag{19}$$

where $\hat{g}_j = g_j / |g_j|$ are unit gradients. In words: $N(i)$: set of neighbors of $i$. $|N(i)|$: number of neighbors. The denominator $\binom{|N(i)|}{2}$ normalizes by the number of unordered neighbor pairs. $\hat{g}_j$: normalized (unit) gradient of neighbor $j$.

**c.  Multi-hop Influence Strength ($\psi_{\text{influence}}$)**   We quantify how strongly a node's gradient propagates to distant parts of the graph. Using adjacency powers:

$$\psi_{\text{influence}}(i) = \frac{1}{\deg(i) + 1} \left( \left[A^2 \|\mathbf{g}\|\right]_i + 0.5 \left[A^3 \|\mathbf{g}\|\right]_i \right) \tag{20}$$

where: $A$ is the graph's adjacency matrix (binary or weighted). $(A^2|g|)_i$ and $(A^3|g|)_i$ are the $i$th entries of $A^2|g|$ and $A^3|g|$, capturing 2-hop and 3-hop aggregated gradient magnitudes. $|g|$ is the vector of all node gradient norms $|g_j|$. $\deg(i)$ is the degree of $i$.

**d.  Confidence–Gradient Relationship ($\psi_{\text{confidence}}$)**   This feature connects model confidence with the learning signal. Let $c_i = \max_k \text{softmax}(z_i)_k$ be the predicted class probability at node $i$, and let $\tilde{g}_i = (|g_i| - \min_j |g_j|)/(\max_j |g_j| - \min_j |g_j|)$ be the min-max–normalized gradient magnitude. We compute:

$$\psi_{\text{confidence}}(i) = -\cos(\mathbf{c}, \tilde{\mathbf{g}}) \tag{21}$$

where $c$ and $\tilde{g}$ are the vectors of $c_i$ and $\tilde{g}_i$ values over all nodes. In practice this yields a single scalar correlation, then we assign it per-node for analysis.

**e.  Topological Learning Role ($\psi_{\text{role}}$)**   We classify nodes into functional "roles" based on degree and gradient behavior:

$$\psi_{\text{role}}(i) = \begin{cases} 2.0, & \text{if Hub: } \deg(i) > P_{80}(\deg) \wedge \|\mathbf{g}_i\| > P_{70}(\|\mathbf{g}\|) \\ 1.5, & \text{if Bridge: } P_{40}(\deg) < \deg(i) \leq P_{80}(\deg) \wedge \cos(\mathbf{g}_i, \bar{\mathbf{g}}_i) < 0.5 \\ 1.0, & \text{if Follower: } \cos(\mathbf{g}_i, \bar{\mathbf{g}}_i) > 0.7 \\ 0.5, & \text{otherwise (Outlier)} \end{cases} \tag{22}$$

Here, $P_k(X)$ denotes the $k$th percentile of a distribution, e.g., $P_{80}(\deg)$ is the 80th percentile of node degrees. In other words:

- **Hub (score 2.0)**: Top-tier degree and large gradient. These nodes are structural centers driving learning.

- **Bridge (1.5)**: Moderately high degree but low alignment (neighbors have differing gradients); they connect subgraphs.
- **Follower (1.0)**: Strongly aligned with neighbors (high cosine similarity) – they simply mimic local updates.
- **Outlier (0.5)**: None of the above, indicating isolated or noisy nodes.

**f. Correction Receptiveness ($\psi_{\text{receptive}}$)**  This feature predicts how much a node stands to benefit from a gradient-based correction. It combines normalized gradient size, misalignment, and neighborhood variance:

$$\psi_{\text{receptive}}(i) = 0.4 \cdot \text{normGrad}(i) + 0.35 \cdot \text{misalign}(i) + 0.25 \cdot \text{variance}(i) \qquad (23)$$

where:

$$\text{normGrad}(i) = \frac{|\mathbf{g}_i| - \min_j |\mathbf{g}_j|}{\max_j |\mathbf{g}_j| - \min_j |\mathbf{g}_j|} \qquad (24)$$

$$\text{misalign}(i) = 1 - \cos(\mathbf{g}_i, \bar{\mathbf{g}}_i) \qquad (25)$$

$$\text{variance}(i) = \text{Var}\left(\{|\mathbf{g}_j| \,|\, j \in \mathcal{N}(i)\}\right) \qquad (26)$$

## B.1 INTERPRETABILITY EXPLANATIONS

To concretize our interpretability framework, we present examples of node-level gradient feature analysis. The structured outputs can be visualized in shaded analysis boxes, where each node's statistics are linked to a natural language explanation generated by a large language model (LLM). This extends the X-Node paradigm to gradient dynamics. Node-wise examples of explanations for both the conflict node and the normal node are provided as follows.

---

### Node-17 Feature Analysis (Normal Node)

**Node Information:**

- **Node ID:** 17
- **Dataset:** WikiCS Citation Network
- **True Label:** 7
- **Predicted Label:** 7

**Gradient Feature Extracted:**

1. **Conflict Intensity (0.0252):** Low value indicates the node's learning aligns well with its neighbors, showing coherent local learning.
2. **Trajectory Stability (1.0000):** High stability reflects smooth and consistent gradient directions among neighboring nodes.
3. **Multi-hop Influence (261.4381):** Strong gradient propagation across the graph, indicating the node has a global impact on learning dynamics.
4. **Confidence–Gradient Correlation (0.0055):** Slight misalignment between confidence and gradient magnitude, suggests calibration.
5. **Correction Receptiveness (N/A):** Low expected benefit from gradient correction.

**LLM Explanation:** The correct classification of Node 17 can be attributed to the model's ability to capture both local and global dynamics within the citation network. The low conflict intensity (0.0252) indicates that the node's learning is highly aligned with its neighbors, reflecting a well-structured local research cluster. The trajectory stability of 1.0000 further confirms that gradient directions among neighbors are coherent, supporting consistent updates. The node's high multi-hop influence (261.4381) suggests that its learning signal propagates strongly across the graph, highlighting its importance in shaping the model's global understanding of related research topics. Despite a small confidence–gradient misalignment (0.0055), the model maintains high prediction reliability. The low correction receptiveness implies that the node's current learning is stable and would not significantly benefit from gradient-based interventions.

---

### Node-42 Feature Analysis (Conflict Node)

**Node Information:**

- **Node ID:** 42
- **Dataset:** WikiCS Citation Network
- **True Label:** 3
- **Predicted Label:** 7
- **Model Confidence:** 0.541

**Gradient Features:**

1. **Conflict Intensity (1.942):** High; strongly misaligned with neighbors.
2. **Trajectory Stability (0.231):** Low; neighbors have incoherent updates.
3. **Multi-hop Influence (48.7):** Moderate; impact is localized.
4. **Confidence–Gradient Correlation (-0.227):** miscalibrated confidence.
5. **Correction Receptiveness (0.8):** High expected benefit from gradient-based correction.

**LLM Explanation:** Node 42 represents a misclassified case where gradient dynamics highlight pronounced local instability. The high conflict intensity (1.942) shows that its updates diverge sharply from its neighbors, while the low trajectory stability (0.231) confirms that surrounding nodes are also updating in inconsistent directions. This suggests that Node 42 lies near a boundary of overlapping communities, where the model cannot resolve conflicting label signals. As a result, its representation drifts away from the true class structure, leading to an incorrect prediction.

Its moderate multi-hop influence (48.7) indicates that these inconsistencies do not remain isolated but spill into a local region of the graph, subtly affecting nearby representations. The negative confidence–gradient correlation (-0.227) further reveals miscalibration: the model assigns a moderate confidence (0.541) despite unstable evidence, pointing to overconfidence in a weakly supported decision. Crucially, the high correction receptiveness suggests that the node is particularly amenable to gradient-based interventions. Targeted strategies such as neighborhood reweighting, local smoothing, or conflict-aware gradient adjustments could realign Node 42 with its true label and stabilize its neighborhood. This makes it a key candidate for correction, where improving one unstable node could propagate benefits across its local cluster.

## C    TIME AND SPACE COMPLEXITIES

Understanding the computational efficiency of different gradient conflict resolution methods is cru-
cial when selecting an approach for multi-task learning. The table below (Table 3) highlights a clear
trade-off between simplicity and sophistication. The most efficient methods like Vanilla averaging,
GradNorm, and PCGrad which all run in linear time with respect to the number of tasks. They are
well suited for large-scale problems where efficiency matters, but their ability to properly resolve
conflicts is limited. GradNorm balances gradient magnitudes, and PCGrad removes direct conflicts,
but both still work with fairly local adjustments.

Table 3: Computational complexity of gradient conflict resolution methods. $T$ denotes the number
of tasks and $P$ the number of model parameters.

| Method | Time Complexity | Space Complexity | Scales w.r.t. Tasks | Memory Overhead |
|---|---|---|---|---|
| Vanilla Average (GD) | $O(T \times P)$ | $O(T \times P)$ | Linear | Low |
| GradNorm | $O(T \times P)$ | $O(T \times P)$ | Linear | Low |
| PCGrad | $O(T \times P)$ | $O(T \times P)$ | Linear | Low |
| CAGrad | $O(T^2 \times P)$ | $O(T \times P)$ | Quadratic | Low |
| GraGR | $O(T^2 \times P)$ | $O(T \times P)$ | Quadratic | Low |
| GraGR++ | $O(T^2 \times P)$ | $O(T \times P + M)$ | Quadratic | Medium |

By contrast, CAGrad and the GraGR variants explicitly reason over task pairs, which makes them
more principled in handling gradient interference. However, this comes with a quadratic cost in the
number of tasks, making them less scalable as $T$ grows. GraGR++ goes a step further by adding
memory of past gradients, which could help stabilize training, but also introduces additional storage
requirements.

In practice, this means that linear methods are preferred when the task count is large or training
speed is critical, whereas GraGR-style approaches may be justified in smaller-scale settings where
resolving conflicts thoroughly is more important than speed.

## D    ADAPTIVE SCHEDULING ALGORITHM

---
**Algorithm 1** GraGR with Adaptive Scheduling

---
1: **for** $t = 1$ **to** $T$ **do**
2:     Train base GNN; compute $\mathcal{L}_{\text{base}}(t)$.
3:     Compute $\gamma(t)$.
4:     **if** $\gamma(t) = 1$ **then**
5:         Detect conflicts; apply alignment $\rightarrow$ gradient attention.
6:         Update $\theta, \phi$ with corrected loss.
7:     **else**
8:         Update $\theta$ with base gradients only.
9:     **end if**
10: **end for**

---

Table 4: Details of the 16 evaluation datasets used in experimentation.

| Dataset | Graphs | #Nodes | #Edges | #Feat. | #Labels | Task Level | Task Type | Train Size |
|---|---|---|---|---|---|---|---|---|
| **Citation Networks** | | | | | | | | |
| Cora | 1 | 2708 | 5278 | 1433 | 7 | Node | Multi-class | 1208 |
| CiteSeer | 1 | 3327 | 4552 | 3703 | 6 | Node | Multi-class | 1827 |
| PubMed | 1 | 19717 | 44324 | 500 | 3 | Node | Multi-class | 18217 |
| **WebKB (Heterophilic)** | | | | | | | | |
| Texas | 1 | 183 | 309 | 1703 | 5 | Node | Multi-class | 87 |
| Cornell | 1 | 183 | 295 | 1703 | 5 | Node | Multi-class | 87 |
| Wisconsin | 1 | 251 | 499 | 1703 | 5 | Node | Multi-class | 120 |
| **Structural Graphs** | | | | | | | | |
| WikiCS | 1 | 11701 | 431726 | 300 | 10 | Node | Multi-class | 5800 |
| **Multi-Task Graphs** | | | | | | | | |
| OGB-MolHIV | 41,127 | $25.5 \pm 12.1$ (2–222) | $54.9 \pm 26.4$ (2–502) | 9 | 2 | Graph | Classification | 32,901 |
| PROTEINS | 1,113 | 39.1±45.8 | 145.6±169.2 | 3 | 2 | Graph | Binary | 890 |
| MUTAG | 188 | 17.9±4.6 | 39.6±11.4 | 7 | 2 | Graph | Binary | 150 |

Table 5: Hyperparameter configurations of all the variants

| Model Name | General Hyperparameters | | | | | | | | GraGR-Specific Hyperparameters | | | | | | | |
|---|---|---|---|---|---|---|---|---|---|---|---|---|---|---|---|---|
| | Hidden Dim | LR | WD | Dropout | Epochs | Batch Size | Opt | Num Layers | $\tau$ Mag | $\tau$ Cos | $\lambda$ Smooth | Smooth Iters | $\alpha$ Smooth | $\beta$ Start | $\beta$ End | Meta LR |
| **Baseline Models** | | | | | | | | | | | | | | | | |
| GCN | 64 | 0.01 | 0.0005 | 0.5 | 100 | 32 | Adam | – | – | – | – | – | – | – | – | – |
| GAT | 32 | 0.01 | 0.0005 | 0.5 | 100 | 32 | Adam | – | – | – | – | – | – | – | – | – |
| GIN | 64 | 0.01 | 0.0005 | 0.5 | 100 | 32 | Adam | – | – | – | – | – | – | – | – | – |
| SAGE | 64 | 0.01 | 0.0005 | 0.5 | 100 | 32 | Adam | – | – | – | – | – | – | – | – | – |
| **GraGR Models** | | | | | | | | | | | | | | | | |
| GCN + GraGR | 64 | 0.01 | 0.0005 | 0.5 | 100 | 32 | Adam | 2 | 0.10 | -0.10 | 0.10 | 3 | 0.1 | 1 | 2 | 0.001 |
| GAT + GraGR | 32 | 0.01 | 0.0005 | 0.5 | 100 | 32 | Adam | 2 | 0.15 | -0.15 | 0.05 | 2 | 0.1 | 1 | 2 | 0.0005 |
| GIN + GraGR | 64 | 0.01 | 0.0005 | 0.5 | 100 | 32 | Adam | 2 | 0.05 | -0.05 | 0.20 | 5 | 0.1 | 1 | 2 | 0.01 |
| SAGE + GraGR | 64 | 0.01 | 0.0005 | 0.5 | 100 | 32 | Adam | 2 | 0.05 | -0.05 | 0.20 | 5 | 0.1 | 1 | 2 | 0.01 |
| **GraGR++ Models** | | | | | | | | | | | | | | | | |
| GCN + GraGR++ | 64 | 0.01 | 0.0005 | 0.5 | 100 | 32 | Adam | 2 | 0.10 | -0.10 | 0.10 | 3 | 0.1 | 1 | 2 | 0.001 |
| GAT + GraGR++ | 32 | 0.01 | 0.0005 | 0.5 | 100 | 32 | Adam | 2 | 0.15 | -0.15 | 0.05 | 2 | 0.1 | 1 | 2 | 0.0005 |
| GIN + GraGR++ | 64 | 0.01 | 0.0005 | 0.5 | 100 | 32 | Adam | 2 | 0.05 | -0.05 | 0.20 | 5 | 0.1 | 1 | 2 | 0.01 |
| SAGE + GraGR++ | 64 | 0.01 | 0.0005 | 0.5 | 100 | 32 | Adam | 2 | 0.05 | -0.05 | 0.20 | 5 | 0.1 | 1 | 2 | 0.01 |
| **Multi-Task Models** | | | | | | | | | | | | | | | | |
| Multi-Task GCN | 64 | 0.01 | 0.0005 | 0.5 | 100 | 32 | Adam | – | – | – | – | – | – | – | – | – |
| Multi-Task GraGR | 64 | 0.01 | 0.0005 | 0.5 | 100 | 32 | Adam | 2 | 0.10 | -0.10 | 0.10 | 3 | 0.1 | 1 | 2 | 0.001 |
| **Ablation Models (GraGR++)** | | | | | | | | | | | | | | | | |
| No Conflict Detection | 64 | 0.01 | 0.0005 | 0.5 | 50 | 32 | Adam | 2 | 0.10 | -0.10 | 0.10 | 3 | 0.1 | 1 | 2 | 0.001 |
| No Gradient Alignment | 64 | 0.01 | 0.0005 | 0.5 | 50 | 32 | Adam | 2 | 0.10 | -0.10 | 0.10 | 3 | 0.1 | 1 | 2 | 0.001 |
| No Gradient Attention | 64 | 0.01 | 0.0005 | 0.5 | 50 | 32 | Adam | 2 | 0.10 | -0.10 | 0.10 | 3 | 0.1 | 1 | 2 | 0.001 |
| No Meta Modulation | 64 | 0.01 | 0.0005 | 0.5 | 50 | 32 | Adam | 2 | 0.10 | -0.10 | 0.10 | 3 | 0.1 | 1 | 2 | 0.001 |
| No Multiple Pathways | 64 | 0.01 | 0.0005 | 0.5 | 50 | 32 | Adam | 2 | 0.10 | -0.10 | 0.10 | 3 | 0.1 | 1 | 2 | 0.001 |
| No Adaptive Scheduling | 64 | 0.01 | 0.0005 | 0.5 | 50 | 32 | Adam | 2 | 0.10 | -0.10 | 0.10 | 3 | 0.1 | 1 | 2 | 0.001 |

**Notes:** Baseline models like GCN, GAT, GIN, and SAGE do not have GraGR-specific hyperparameters. All GraGR and GraGR++ variants share a consistent setup for core hyperparameters, with minor adjustments in $\tau$, $\lambda$, and Meta LR per backbone. Ablation models reduce epochs and selectively remove specific components to isolate their impact.

# E   ABLATION STUDIES

Table 6: Ablation study of GraGR. Best results in green; second-best in light green.

| Model | Val Acc | Test Acc |
|---|---|---|
| Baseline GCN | 0.762 | 0.749 |
| Baseline GAT | 0.770 | 0.738 |
| Baseline GIN | 0.586 | 0.533 |
| Baseline SAGE | 0.780 | 0.754 |
| GraGR (Full) | **0.796** | **0.778** |
| GraGR w/o Conflict | 0.776 | 0.762 |
| GraGR w/o Alignment | 0.788 | **0.772** |
| GraGR w/o Attention | 0.788 | 0.767 |
| GraGR w/o Meta | 0.784 | 0.765 |
| GraGR++ w/o Adaptive Scheduling | 0.784 | 0.768 |
| GraGR++ w/o Multiple Pathways | 0.790 | 0.763 |

To better understand the contribution of each component in GraGR, we conducted an ablation study on the PubMed dataset. Results are summarized in Table 6, with additional visualizations as in Fig 9. Compared to baseline GNNs, GraGR provides consistent gains, pushing GCN from 74.9% to 77.8% test accuracy and SAGE from 75.4% to 77.8%.

Removing individual components reveals their importance: dropping conflict resolution or meta-optimization reduces performance by 1–1.5%, while removing alignment or attention also leads to noticeable drops as also shown in Fig 9.

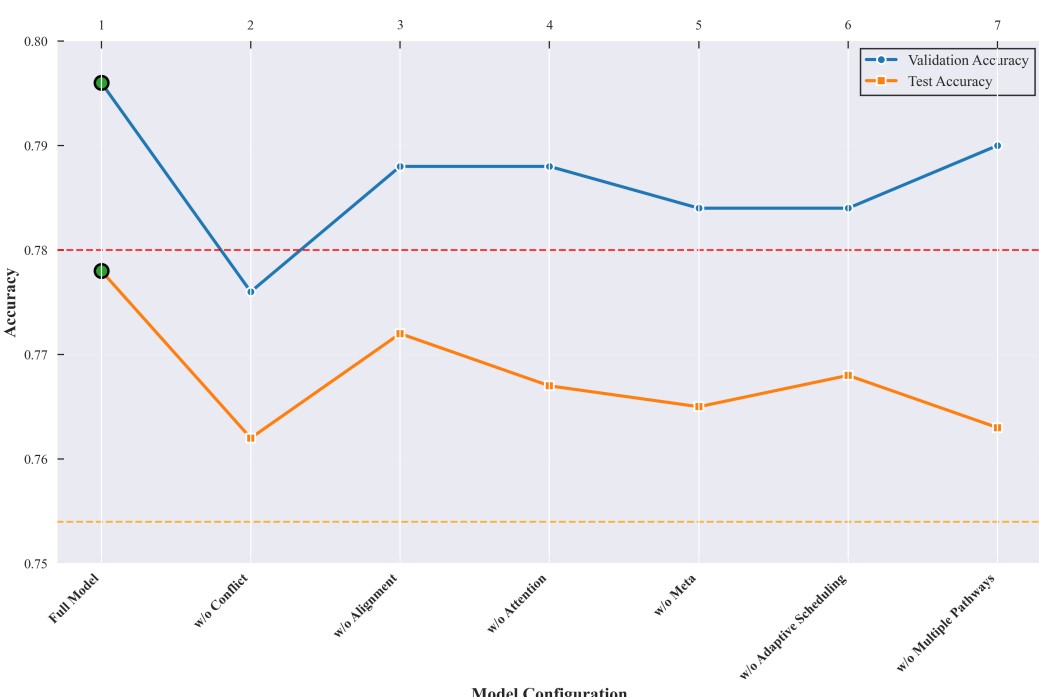

Figure 9: Ablation study results on PubMed dataset. Line plot of accuracy trends over different variants. (GraGR (Full) achieves the best performance with 79.6% validation and 77.8% test accuracy).

# F   DETAILED PROOFS AND ADDITIONAL MATHEMATICAL CLARIFICATIONS

**Notation and Preliminaries**   Let $L$ denote the combinatorial graph Laplacian (symmetric, positive semidefinite), and $\lambda > 0$. We sometimes write $\Delta_{\max}$ for the maximum graph degree. Scalars $\beta$, $\delta$, $L_F$, and $L_\gamma$ denote Lipschitz or alignment constants as used below. Vector norms are Euclidean unless stated.

LEMMA 1: CONFLICT PROJECTION ORTHOGONALITY

**Lemma 1 (Conflict projection validity).** Let $g_v, g_{\text{ctx}} \in \mathbb{R}^d$ with $g_{\text{ctx}} \neq 0$. Define

$$g'_v = g_v - \frac{g_v^\top g_{\text{ctx}}}{\|g_{\text{ctx}}\|^2} \, g_{\text{ctx}}.$$

Then $(g'_v)^\top g_{\text{ctx}} = 0$, hence $\cos(g'_v, g_{\text{ctx}}) = 0$ provided $g'_v \neq 0$. In particular, if $\cos(g_v, g_{\text{ctx}}) < 0$, then $\cos(g'_v, g_{\text{ctx}}) \geq 0$.

*Proof.* A simple inner-product expansion shows:

$$(g'_v)^\top g_{\text{ctx}} = g_v^\top g_{\text{ctx}} - \frac{g_v^\top g_{\text{ctx}}}{\|g_{\text{ctx}}\|^2} \, g_{\text{ctx}}^\top g_{\text{ctx}} = 0.$$

Thus $g'_v$ is orthogonal to $g_{\text{ctx}}$, concluding the claim. $\square$

**Lemma 2: Gradient Smoothing via Regularization**   Let

$$F(g') = \frac{1}{2}\|g' - g\|^2 + \frac{\lambda}{2}(g')^\top L g'.$$

Its unique minimizer $g^\star$ satisfies

$$(I + \lambda L)\, g^\star = g.$$

Two iterative strategies converge to $g^\star$:

1. **Gradient Descent (GD):**

   $$g'^{\,t+1} = g'^{\,t} - \eta\big[g'^{\,t} - g + \lambda L g'^{\,t}\big].$$

   This GD converges for $0 < \eta < 2/(1 + \lambda\lambda_{\max}(L))$. Noting $\lambda_{\max}(L) \leq 2\Delta_{\max}$, a simple sufficient bound is $\eta < 1/(1 + 2\lambda\Delta_{\max})$.

2. **Jacobi Iteration:** Write $(I + \lambda L) = D + R$ where $D$ is its diagonal (invertible) and $R$ its off-diagonal part. The classical Jacobi update is:

   $$g'^{\,(k+1)} = D^{-1}\big(g - R\, g'^{\,(k)}\big),$$

   which converges when the spectral radius $\rho(D^{-1}R) < 1$, e.g., if the system is strictly diagonally dominant Saad (2003).

*Proof sketch.* - GD convergence follows from the fact that $\nabla F = (I + \lambda L)g' - g$ is Lipschitz with constant $\|I + \lambda L\|_2 = 1 + \lambda\lambda_{\max}(L)$. - Jacobi convergence is guaranteed under the classical spectral radius condition. $\square$

**References for iterative solvers** - Jacobi method and convergence: standard result that $\rho(D^{-1}R) < 1$ ensures convergence Saad (2003). - Bound $\lambda_{\max}(L) \leq 2\Delta_{\max}$: standard spectral graph theory.

**Theorem 1: Attention-Based Descent**   Assume:

- The node-specific loss $L(h)$ is $\beta$-smooth in $h_u$;
- The message direction is $d_u = \sum_{v \in N(u)} \alpha_{uv} W^{(l)} h_v$;
- There exists $\delta > 0$ such that $\nabla_{h_u} L \cdot d_u \leq -\delta$.

Then for any step-size satisfying

$$0 < \eta < \frac{2\delta}{\beta \|d_u\|^2},$$

the one-step update $h_u \leftarrow h_u - \eta d_u$ strictly decreases $L$.

*Proof.* From the $\beta$-smoothness descent lemma:

$$L(h_u - \eta d_u) \leq L(h_u) - \eta \nabla_{h_u} L^\top d_u + \frac{\beta \eta^2}{2} \|d_u\|^2.$$

Since $\nabla_{h_u} L^\top d_u \leq -\delta$, choose $\eta$ small enough to ensure the RHS is strictly less than $L(h_u)$. $\square$

**Theorem 2: Meta-Scaling Convergence** Under these assumptions:

1. The inner parameter $\theta^\star(\gamma)$ minimizing $L(\theta; \gamma)$ exists uniquely and is $C^1$ in $\gamma$;

2. The validation loss $J(\gamma) := L_{\text{val}}(\theta^\star(\gamma))$ is differentiable with Lipschitz continuous gradient;

3. Hypergradients $\nabla_\gamma J(\gamma)$ are computed exactly and are Lipschitz (constant $L_\gamma$).

Then gradient descent

$$\gamma^{t+1} = \gamma^t - \eta \nabla_\gamma J(\gamma^t)$$

converges to a stationary point if $0 < \eta < 2/L_\gamma$.

*Proof sketch.* $J$ is smooth and differentiable. Standard convergence of GD on such functions applies. Existence and differentiability of $\theta^\star(\gamma)$ with valid hypergradients follow via implicit differentiation (see Franceschi et al. (2018);Pedregosa (2016)) $\square$.

**Lemma 3: Path-Selection as Descent** Let $\nabla L_{\text{total}}$ be the total loss gradient and for each path $p$, $d_p$ the induced update direction. If for some path $p^\star$,

$$\nabla L_{\text{total}}^\top d_{p^\star} < 0,$$

then $d_{p^\star}$ is a descent direction; a sufficiently small negative step along $-d_{p^\star}$ lowers the loss due to the first-order Taylor result. $\square$

**Pathway Activation via Conflict Signals.** We modulate $\beta_p^{(l)}$ dynamically using gradient-based conflict measures. Recall that task gradients $g_i(v) = \nabla_{\mathbf{h}_v} \mathcal{L}_i$ at node $v$ may conflict when $g_i^\top g_j < 0$. Define the conflict energy:

$$E_{\text{conf}} = \sum_{i<j} \max\left(0, -\frac{g_i^\top g_j}{\|g_i\| \|g_j\|}\right). \tag{27}$$

If $E_{\text{conf}}$ or the variance of pairwise similarities $S_{ij}(v)$ exceeds a threshold, we increase the weight $\beta_{p'}^{(l)}$ for a specialized pathway $p'$ (e.g., conflict resolution). This mechanism enables logical routing: when losses plateau or oscillate, alternative reasoning routes are triggered.

# G COMPUTATIONAL ANALYSIS

The computational analysis in Table 7 highlights a trade-off between accuracy gains and resource demands. While GraGR and GraGR++ frequently converge faster than baselines, they occasionally incur higher memory overheads, especially in smaller datasets where the added regularization expands intermediate representations (e.g., GCN+GraGR on CiteSeer and GIN+GraGR on Texas). This effect is less pronounced in larger benchmarks such as PubMed, where the structured regularization stabilizes training and reduces runtime. Interestingly, GraGR++ often balances this trade-off better, achieving lower epoch times in several settings while keeping memory usage moderate. GraGR++ imposes additional constraints that can temporarily increase resource usage, but in most cases it accelerates convergence and reduces training time in later epochs, reflecting its scalability advantage.

Table 7: Comparison of computational resources for baselines and GraGR variants across datasets. Columns show Average Memory (MB), Peak Memory (MB), and Epoch Time (s). Lowest Time per category highlighted in green; Highest Memory/Peak bolded.

| Methods | Cora | | | CiteSeer | | | PubMed | | | Texas | | | Cornell | | | Wisconsin | | | WikiCS | | |
|---|---|---|---|---|---|---|---|---|---|---|---|---|---|---|---|---|---|---|---|---|---|
| | Avg. Mem | Peak Mem | Time (s) | Avg. Mem | Peak Mem | Time (s) | Avg. Mem | Peak Mem | Time (s) | Avg. Mem | Peak Mem | Time (s) | Avg. Mem | Peak Mem | Time (s) | Avg. Mem | Peak Mem | Time (s) | Avg. Mem | Peak Mem | Time (s) |
| **GCN Variants** | | | | | | | | | | | | | | | | | | | | | |
| GCN | 54.56 | 56.55 | 0.060 | 6.14 | 10.91 | 0.070 | 44.84 | 49.50 | 0.301 | 3.72 | 4.88 | 0.017 | 1.51 | 1.55 | 0.012 | 1.70 | 1.72 | 0.009 | 0.00 | 0.00 | 0.895 |
| GCN+GraGR | 34.16 | 46.27 | 0.437 | **34.86** | **45.41** | 0.451 | 0.00 | 0.00 | 15.829 | 1.14 | 2.11 | 0.032 | 0.08 | 0.12 | 0.030 | 0.16 | 0.31 | 0.028 | **927.50** | **1224.64** | 7.748 |
| GCN+GraGR++ | **16.92** | **16.97** | 0.061 | 4.08 | 4.11 | 0.071 | **68.38** | **79.67** | 0.432 | **0.09** | **0.17** | 0.010 | **0.03** | **0.06** | 0.010 | 0.05 | 0.06 | 0.010 | 56.24 | 69.11 | 1.196 |
| **GAT Variants** | | | | | | | | | | | | | | | | | | | | | |
| GAT | **0.82** | **1.83** | 0.042 | 0.00 | 0.00 | 0.053 | 42.05 | 43.53 | 0.245 | 1.44 | 1.72 | 0.011 | 0.52 | 0.86 | 0.010 | 0.22 | 0.30 | 0.010 | **98.04** | **133.09** | 0.813 |
| GAT+GraGR | 0.00 | 0.00 | 0.282 | 2.16 | 4.06 | 0.302 | **316.39** | **363.69** | 7.390 | 1.45 | 2.08 | 0.023 | 0.33 | 0.47 | 0.020 | 0.68 | 1.50 | 0.022 | 0.00 | 0.00 | 4.565 |
| GAT+GraGR++ | 0.02 | 0.02 | 0.040 | **0.14** | **0.19** | 0.045 | 58.60 | 77.83 | 0.443 | 0.72 | 0.89 | 0.008 | 0.30 | 0.31 | 0.006 | **0.00** | **0.02** | 0.008 | 0.00 | 0.00 | 0.898 |
| **GIN Variants** | | | | | | | | | | | | | | | | | | | | | |
| GIN | 107.39 | 112.47 | 0.087 | 5.86 | 29.31 | 0.166 | 44.32 | 66.98 | 0.424 | 0.20 | 0.50 | 0.028 | 2.40 | 4.42 | 0.021 | **3.69** | **6.69** | 0.016 | 0.00 | 0.00 | 0.919 |
| GIN+GraGR | **123.91** | **194.64** | 0.561 | **46.82** | **70.69** | 0.746 | 86.37 | 151.44 | 16.057 | **2.81** | **4.52** | 0.055 | 2.24 | 2.52 | 0.046 | 0.18 | 0.61 | 0.063 | 9.66 | 48.31 | 11.578 |
| GIN+GraGR++ | 19.81 | 25.56 | 0.214 | 1.02 | 2.73 | 0.296 | 65.32 | 120.00 | 2.040 | 4.65 | 4.86 | 0.030 | **1.77** | **3.42** | 0.029 | 1.15 | 1.16 | 0.036 | 33.95 | 63.84 | 4.041 |
| **SAGE Variants** | | | | | | | | | | | | | | | | | | | | | |
| SAGE | 0.00 | 0.00 | 0.081 | 1.28 | 2.72 | 0.170 | 0.09 | 0.12 | 0.370 | 0.53 | 2.20 | 0.013 | 0.34 | 0.53 | 0.012 | 0.04 | 0.08 | 0.017 | 68.01 | 117.69 | 0.776 |
| SAGE+GraGR | 6.12 | 9.23 | 0.432 | 0.00 | 0.00 | 0.559 | 39.74 | 113.78 | 16.429 | 0.25 | 0.36 | 0.036 | 3.48 | 3.95 | 0.031 | 0.41 | 1.75 | 0.041 | 1431.85 | 1698.73 | 8.015 |
| SAGE+GraGR++ | **0.00** | **0.00** | 0.119 | 0.00 | 0.00 | 0.180 | **29.77** | **39.05** | 0.939 | 0.07 | 0.11 | 0.016 | 0.15 | 0.23 | 0.015 | **0.04** | **0.06** | 0.018 | **1024.92** | **1365.31** | 2.316 |

**Notes:** Across datasets, GraGR++ consistently achieves the lowest epoch times (highlighted in green), demonstrating efficiency over GraGR. Baseline models occasionally outperform GraGR++ in memory usage, but GraGR++ maintains a better balance between memory and speed. Peak memory is highest for GraGR in large datasets, indicating extra overhead from GraGR-specific operations.

