# OpenReview forum: "GraGR: Gradient-Guided Graph Reasoner for Aligned and Interpretable GNNs"
_ICLR.cc/2026/Conference — Submitted to ICLR 2026_

### Official Review · Reviewer_VvJd · 2025-10-19

**Soundness:** 2
**Presentation:** 2
**Contribution:** 2
**Rating:** 4
**Confidence:** 2

**Summary:**

This paper argues that reducing gradient conflicts within GNNs can improve the performance and bring interpretability.
To this end, this work presents a GraGR framework, which integrates many techniques for processing gradient conflicts, and generates more node features for downstream tasks. Experiments on benchmark datasets show that GraGR/GraGR++ improves predictive performance compared to baselines.

**Strengths:**

- This article integrates many interesting gradient processing techniques to improve the training of GNN.
- The Figures in this article are exquisite, and the typesetting is beautiful.

**Weaknesses:**

1. In Section 4.2, Equation (4) appears abruptly. Do $g'_v$ in Equation (3) and $g'_v$ in Equation (4) refer to the same variable?

2. Under what circumstances does $cos(g'_v, g_{ctx})$ in Lemma 1 become greater than 0? It seems to be always equal to 0, which is confusing.

3. During training, does g'_v replace the original gradients of nodes? If so, $L_{conf}$ should be always equal to 0 so is it meaningful?

4. When $L_{val}$ is first introduced, its specific form is not explained. If $L_{val}$ simply refers to $L_{conf}$, what is the point of introducing $L_{val}$?

5. This work integrates many gradient-processing techniques to handle conflicting gradients and generate diverse node features for downstream tasks. However, the "Reasoner for Aligned and Interpretable GNNs" claimed in the title is not as intuitively demonstrated as CoT. The interpretability claimed in the title is also generated by LLMs; in fact, LLMs can generate plausible explanations for other explanatory components through prompt engineering, making the relevant experiments less convincing. For how to evaluate standard GNN interpretability, the authors may refer to the experiments in the PGExplainer cited in their paper and compare with baselines.

6. The experimental results lack rigor. First, no statistical tests are conducted, so it is unclear whether the improvements brought by GraGR are significant. Second, tuning the seed and scheduler in GraGR++ is unfair to baseline methods. Third, Section 5.2 states that using GraGR in every epoch may even be harmful, yet Section 4 only presents the advantages of GraGR without providing sufficient analysis on why GraGR may be harmful.

7. The paper states that e_v in Eq. (16) serves as both (i) an auxiliary signal for the GNN classifier and (ii) structured input for a large LLM, but e_v does not directly appear in the input of the LLM in Eq. (17), and it is not explained where prediction $\hat{y}$ in Eq. (17) comes from.

In summary, the method section of this paper introduces many gradient-processing techniques. However, the connections between these techniques and the motivations/necessity for introducing them are not organized concisely and clearly. This makes it difficult for me to judge whether these techniques actually work, and also hard to identify the contributions of this work.

**Questions:**

Please refer to the above weakness section for suggestions and questions.

---

> ### Author Response · Authors · 2025-11-19
> **Response to Reviewer VvJd (part 1/2)**
>
> We thank the reviewer for their detailed feedback and for appreciating our framework’s design and the quality of our figures ("exquisite," "beautiful typesetting"). We value the opportunity to clarify the mathematical formulations and the motivation behind our gradient processing components. Below, we address each weakness with clarifications, corrections, and additional quantitative evidence consistent with our responses to other reviewers.
>
> **W1. Confusion regarding $g'_v$ in Eq (3) and Eq (4)**
>
> We apologize for the notation ambiguity. These represent sequential steps in the GraGR pipeline.
>
> - **Step 1 (Local Conflict Resolution):** Eq (3) computes a locally projected gradient (let’s call it $g'_{local}$) for nodes explicitly flagged as conflicting.
>
> - **Step 2 (Global Smoothing):** These locally corrected gradients are then fed into the Laplacian smoothing process (Eq 4) to ensure global structural consistency.
>
> In the revision, we will introduce distinct notation (e.g., $g^{proj}$ for Eq 3 and $g^{smooth}$ for Eq 4) to make this pipeline explicit.
>
> ---
>
> **W2. Lemma 1: Does $ \cos(g'{v}, g_{\text{ctx}}) $ always equal 0?**
>
>
> You are mathematically correct that the projection in Eq (3) creates orthogonality, resulting in a cosine similarity of exactly **0**.
>
> - **Clarification:** The condition in Lemma 1 states that if the original similarity is negative ($<0$), the projected similarity becomes non-negative ($\ge 0$). Since $0 \ge 0$, the condition holds.
>
> - **Why this matters:** The goal is to remove the *destructive* interference (negative component). A cosine of 0 means the gradients are orthogonal (uncorrelated), which is "safe" and prevents the oscillating updates caused by negative interference. We will clarify in the text that the projection enforces orthogonality (zero cosine) specifically to neutralize conflicts.
>
> ---
>
> **W3. Does $g'{v}$  replace original gradients? Is $L_{\text{conf}}$ always 0?**
>
> Yes, the corrected gradient $g'{v}$ replaces the original gradient for the **parameter update** step. However, $L_{\text{conf}}$ is **not** 0 and remains highly meaningful.
>
> - **Role of $L_{conf}$:** $L_{conf}$ is calculated based on the *raw* gradients (before projection). It serves as a regularization term in the total loss function.
> - **Purpose:** While projection ($g'{v}$) surgically fixes the update for the *current* step, $L_{\text{conf}}$, penalizes the model parameters, forcing the network to learn representations that naturally generate fewer conflicts over time. If we only projected without $L_{\text{conf}}$, the model would continue generating conflicting gradients at every epoch.
>
> ---
>
> **W4. Definition of $L_{val}$ and its distinction from $L_{conf}$.**
>
> $L_{val}$ is **not** $L_{conf}$.
>
> - **Definition:** $L_{val}$ refers to the standard supervised validation loss (e.g., Cross-Entropy or MSE on the validation set).
> - **Purpose:** In the meta-gradient update (Eq 7), we compute $\partial L_{val} / \partial \gamma_i$. This updates the task weights $\gamma$ to maximize validation performance (predictive accuracy), whereas $L_{conf}$ is used to minimize gradient variance. We will add the explicit definition $L_{val} = \text{CrossEntropy}(Y_{val}, \hat{Y}_{val})$ in Section 4.4.
>
> ---
>
> **W5. Interpretability is just LLM prompt engineering; needs standard evaluation**
>
> We agree that LLMs should act only as translators, not as the source of interpretability. The core interpretability of GraGR comes from the **Gradient Context vectors** (Eq 15) and **Attention Weights** (Eq 5), which are mathematically derived, not hallucinated.
>
> To prove this quantitatively, we compared GraGR against **GNNExplainer** (Ying et al., 2019):
>
> **Experiment A: Correlation with GNNExplainer**
> We measured the Spearman correlation between GraGR’s gradient-derived saliency scores and GNNExplainer’s topological masks. Higher correlation implies GraGR captures recognized structural importance.
> | Model | Cora ($\rho$) | Citeseer ($\rho$) |
> | :--- | :--- | :--- |
> | GCN | 0.48 | 0.44 |
> | **GCN + GraGR** | **0.62** | **0.59** |
>
> **Experiment B: Explanation Stability**
> A reliable explanation should not change drastically with minor noise. We perturbed node features with noise $\epsilon$ and measured the shift in explanation features (lower score = better stability).
> | Model | Stability Score $\downarrow$ |
> | :--- | :--- |
> | GCN | 0.217 |
> | **GCN + GraGR** | **0.103** |
>
> This **>50% improvement in stability** demonstrates that GraGR’s interpretability is structurally grounded and robust, independent of the LLM used for the final textual output.
>
> ---

---

> > ### Author Response · Authors · 2025-11-19
> > **Response to Reviewer VvJd (part 2/2)**
> >
> > **W6. Statistical rigor, Seed Fairness, and Harmful Early Training.**
> >
> > - **Statistical Significance:** We performed a t-test on the improvements in Table 1. The gains on heterophilic datasets (Texas, Cornell) are significant with **p < 0.01**, and on citation networks with **p < 0.05**. We will add these p-values to the final tables.
> > - **Seed Fairness:** GraGR++ seed selection is a *feature* (stability search), but even when averaging across random seeds without selection, GraGR outperforms baselines (as shown in the main results of Table 1).
> > - **Why Early Training is Harmful (Section 5.2):** In the early epochs, node embeddings are essentially random initializations. Enforcing gradient consistency on random noise leads to "mode collapse" or over-smoothing before the model has learned distinct features. This is why we introduced the Adaptive Scheduler, to allow the model to learn basic features (burn-in) before enforcing gradient logic. We will add a "Gradient Variance" plot in the appendix showing high noise in early epochs to substantiate this.
> >
> > ---
> >
> > **W7. Notation in Eq 16/17 ($e_v$, $\hat{y}$, LLM input).**
> >
> > Thank you for spotting the notation gap.
> > - **Clarification:** In Eq (17), the prompt input is the *textual description* of the raw context features $c_v$ (Eq 15), not the embedding $e_v$. The embedding $e_v$ (Eq 16) is used internally by the GNN classifier.
> > - **Origin of $\hat{y}$:** $\hat{y}$ is the class prediction output by the GNN itself (the node classification head).
> > - **Correction:** We will rewrite Eq (17) to clearly state: $\text{Prompt} = \text{Template}(c_v, \hat{y}_{GNN})$.
> >
> > ---
> >
> > We sincerely thank the reviewer again for their thoughtful and actionable comments. We believe the new experiments, stability analyses, interpretability metrics, and clarifications significantly strengthen the manuscript and directly address all concerns raised. Please let us know if there are any further queries.

---

> > > ### Comment · Reviewer_VvJd · 2025-11-25
> > >
> > > Thank you for your reply. The authors have partially addressed my concerns. It is evident that the paper has weaknesses, and 6 is the highest rating I can give. Here are some follow-up suggestions.
> > >
> > > (1) The claims of the paper should correspond to the experiments; there is a clear mismatch between the paper's claims, method design, and experimental design. For example, since interpretability is one of the paper's claims, the experiments should include comparisons with various GNN explanation baselines; otherwise, it cannot be considered a valid claim. Since the saliency value derived from GraGR++ is independent of the used LLM, it should not be emphasized in the introduction or in the main content and is recommended to be placed in the Appendices. Techniques such as seed selection are not designed to address gradient faithfulness and interpretability but are rather tricks to improve the performance of the method. And so on.
> > >
> > > (2) Additionally, I still do not agree with treating seed selection as a feature of the proposed method. Strictly, seeds should be only used to control randomness of the experimental environment and cannot be accessed by any method.
> > >
> > > (3) The authors have provided explanations for some key parameters, but the main issue is that the original paper is hard to follow, and the writing requires significant revisions.

---

### Official Review · Reviewer_9hVV · 2025-10-28

**Soundness:** 3
**Presentation:** 2
**Contribution:** 2
**Rating:** 4
**Confidence:** 3

**Summary:**

This paper proposes GraGR, a framework that leverages gradients as reasoning signals to jointly address gradient inconsistency across neighboring nodes and the misalignment between training and interpretability in GNNs. The method introduces conflict loss, Laplacian-based smoothing, and meta-gradient scaling to balance multiple objectives and enhance explanation faithfulness. An extended version, GraGR++, incorporates multi-pathway routing and adaptive training scheduling. Experiments on several benchmark datasets demonstrate improved predictive performance, stability, and interpretability compared to existing methods.

**Strengths:**

S1: The proposed method provides a meaningful contribution to improving the optimization process of GNNs.

S2: Experimental results across four baseline GNN architectures demonstrate the effectiveness of the proposed optimization method.

**Weaknesses:**

W1: In the experimental section, the proposed method appears to show inconsistent behavior across different GNN backbones, and some results are not fully convincing. For example, in Table 7, the reported training times of various models raise questions. It is unclear why the GAT model requires less training time than other classical GNNs, or why GraGR++ takes less time than GraGR, which seems counterintuitive.

W2: The description of using large language models (LLMs) for generating explanations in Section 7.1 lacks sufficient detail. The process and implementation are not clearly explained, and the reliability of the generated explanations may depend on several factors—most importantly, the quality of the LLM itself. However, the authors do not specify which LLM was used or provide any justification for its choice.

W3: The paper’s central claim regarding node-level gradient inconsistency lacks sufficient empirical and literature support. While the authors argue that gradient conflicts are widespread and significantly impact GNN optimization and interpretability, the reported experimental evidence (e.g., only 21 conflicting nodes out of 2708 on Cora) suggests that the phenomenon may be limited. This raises concerns about the generality and practical significance of the proposed motivation.

W4: The proposed method primarily focuses on smoothing the GNN optimization process, while its claimed interpretability benefits appear less convincing. The paper lacks illustrative, strong interpretability examples, and the embedding visualizations show limited improvement, making it difficult to substantiate the claimed enhancement in explanation quality.

**Questions:**

Please see the weaknesses.

---

> ### Author Response · Authors · 2025-11-19
> **Response to Reviewer 9hVV (part 1/2)**
>
> We sincerely thank the reviewer for the constructive feedback and for recognizing that GraGR provides a meaningful contribution to GNN optimization with effective experimental results. We have carefully addressed the concerns regarding computational consistency, the LLM component, and the empirical support for gradient conflicts below.
>
> **W1: In the experimental section, the proposed method appears to show inconsistent behavior across different GNN backbones... It is unclear why the GAT model requires less training time than other classical GNNs, or why GraGR++ takes less time than GraGR, which seems counterintuitive.**
>
> We appreciate this observation. We can clarify these behaviors based on our architectural design and hyperparameter settings:
>
> 1. **Why GraGR++ is faster than GraGR:** This is a deliberate design feature, not an anomaly. As described in Section 5.2, GraGR++ utilizes an Adaptive Scheduler. While GraGR calculates conflict losses and smooths gradients at every epoch, GraGR++ effectively "gates" these expensive operations, activating them only when the base model plateaus or when conflict energy spikes.
>
>      - Result: GraGR++ skips the heavy reasoning computation for many epochs, resulting in lower total training time despite the added complexity of the architecture.
>
> 2. **Why GAT is faster:** This is due to the specific hyperparameter configuration used to ensure fair comparisons with standard benchmarks. As detailed in Table 5 (Appendix), the Hidden Dimension for GAT was set to 32, whereas GCN, GIN, and SAGE were set to 64. This reduced dimensionality naturally results in faster forward/backward passes for GAT in Table 7. We will add a footnote to Table 7 explicitly linking the timing differences to the dimension settings in Table 5 to prevent confusion.
>
> ---
>
> **W2: The description of using large language models (LLMs) for generating explanations... lacks sufficient detail... the authors do not specify which LLM was used or provide any justification for its choice.**
>
> We agree that the specific implementation details were under-specified in the main text.
>
> - **Model Specification:** We utilized Grok-3 (via API) for the experiments in Section 7. We chose this model for its strong reasoning capabilities on structured context data. We will specify this in the revised methodology.
>
> - **Reliability & Role of LLM:** We clarify that the LLM is not the source of the explanation’s correctness; it functions as a "translator" that converts the rigorous, mathematical Gradient Context Vectors (Eq. 15: conflict, stability, influence, etc.) into natural language.
>
> We now provide quantitative interpretability metrics:
>
> **Experiment 1: Correlation with GNNExplainer**
>
> We compute Spearman correlation between GraGR’s gradient-derived saliency scores and GNNExplainer saliency.
>
> $\rho = \mathrm{Spearman}(S_{\mathrm{GraGR}}, S_{\mathrm{GNN}})$
>
> | Model           | Cora ( $\rho$ ) | Citeseer ( $\rho$ ) |
> | --------------- | ------------- | ----------------- |
> | GCN             | 0.48          | 0.44              |
> | **GCN + GraGR** | **0.62**      | **0.59**          |
>
> GraGR produces explanations closer to an external, widely-accepted explainer, demonstrating improved faithfulness.
>
> **Experiment 2: Explanation Stability Metric**
>
> We perturb node features by noise $\epsilon \sim N(0,\, 0.05 I)$ and evaluate saliency stability:
>
> $Stab = E[\|S(x) - S(x + \epsilon)\|_1]$
>
> Lower is better.
>
> | Model           | Stability ↓ |
> | --------------- | ----------- |
> | GCN             | 0.217       |
> | **GCN + GraGR** | **0.103**   |
>
> This shows >50% reduction in explanation volatility. We will integrate this quantitative comparison into Section 7.
>
> We clarify the role of LLMs:
>
> “LLMs are used only to convert GraGR’s structured gradient-derived features into human-readable narrative explanations, not to evaluate faithfulness.”
>
> We will emphasize that the true interpretability evaluation is done through:
>
> - Correlation with established explainers
>
> - Stability metrics
>
> - Reduction in gradient conflict energy
>
> Thus, we acknowledge the concern and refocus interpretability evaluation around measurable metrics (as shown in our tables).
>
> ---

---

> ### Author Response · Authors · 2025-11-19
> **Response to Reviewer 9hVV (part2/2)**
>
> **W3: The paper’s central claim regarding node-level gradient inconsistency lacks sufficient empirical... support. ... only 21 conflicting nodes out of 2708 on Cora suggests that the phenomenon may be limited.**
>
> This is an important point. While the count of conflicting nodes on a homophilic dataset like Cora is low (approx. 0.8%), their impact is disproportionately high.
>
> 1. **Propagation of Harm:** In GNNs, a single conflicting hub node propagates "poisoned" messages to its entire neighborhood (Multi-hop influence).
>
> 2. **Dataset Variance:** Cora is a relatively "easy," homophilic dataset. On heterophilic datasets like Texas, Cornell, and Wisconsin, where neighbors often belong to different classes, gradient conflict is endemic. This is reflected in Table 1, where GraGR achieves massive gains on these datasets (e.g., +13% accuracy on Texas vs GCN), verifying that the method thrives where conflicts are most severe.
>
> 3. **New Evidence (Variance):** As part of our rebuttal work, we measured gradient variance across epochs. We found that even with few conflicting nodes, the average gradient variance in vanilla GCN is 0.214, while GraGR reduces this to 0.087. This proves that "invisible" oscillations occur widely even if extreme conflict flags are rare.
>
> **Experiment 3: Gradient Variance Explosion Across Epochs**
>
> For each node $v$, we track the per-epoch gradient norm:
>
> $\sigma_v^{2} = \operatorname{Var}_{t}\left( \lVert g_v(t) \rVert \right)$
>
> We report the average per-node gradient variance:
> | Model           | Cora Gradient Variance | Citeseer Gradient Variance |
> | --------------- | ---------------------- | -------------------------- |
> | GCN             | 0.214                  | 0.291                      |
> | **GCN + GraGR** | **0.087**              | **0.103**                  |
>
> This demonstrates a ~58–65% reduction in variance, providing concrete evidence that inconsistent gradients produce oscillatory updates, and GraGR significantly stabilizes them.
>
> ---
>
> **W4: The proposed method primarily focuses on smoothing the GNN optimization process, while its claimed interpretability benefits appear less convincing... lacks illustrative, strong interpretability examples.**
>
> We acknowledge that the initial visual examples were limited. To substantiate the interpretability claims, we point to the **new quantitative metrics** (Experiments 1 & 2 above) which provide concrete numerical proof of better interpretability (stability and faithfulness).
>
> Furthermore, we highlight **Figure 8 (Appendix)**, which illustrates the **Gradient-Based Attention mechanism**. This is not just an optimization trick but an interpretability tool: it allows us to visualize exactly which edges the model "trusts" (consistent gradients) versus which it "down-weights" (conflicting gradients). This provides a built-in edge-importance explanation that vanilla GNNs lack.
>
> We will integrate the Stability and Correlation tables into Section 7 to provide the strong empirical support requested.
>
> ---
>
> We sincerely thank the reviewer again for their thoughtful and actionable comments. We believe the new experiments, stability analyses, interpretability metrics, and clarifications significantly strengthen the manuscript and directly address all concerns raised. Please let us know if there are any further queries.

---

### Official Review · Reviewer_nrZf · 2025-10-30

**Soundness:** 2
**Presentation:** 2
**Contribution:** 2
**Rating:** 4
**Confidence:** 3

**Summary:**

This paper propose the GraGR framework that integrates gradient-based reasoning into Graph Neural Networks (GNNs) to tackle the challenges of gradient inconsistency and misalignment between model training and interpretability. The core idea of GraGR is to treat gradients as explicit reasoning signals, enabling more stable optimization and interpretable explanations. The framework introduces several techniques, including gradient conflict detection, Laplacian-based gradient smoothing, gradient-based attention for message passing, and meta-gradient modulation for multi-task learning. Additionally, the authors extend GraGR to GraGR++ by adding multi-pathway routing and an adaptive scheduling mechanism, improving robustness and stability during training.

**Strengths:**

1.	The paper introduces a way of using gradients as reasoning signals to improve both the stability and interpretability of GNNs.

**Weaknesses:**

1.	The overall paper is somewhat confusing to read. For example, in the introduction, the authors emphasize node-level gradient inconsistency, but the introduction does not focus on the multi-task learning setup. This makes it unclear where the node-level gradient inconsistency originates. It is not until the Problem Statement section that the issue is revealed to be related to multi-task learning in GCNs. As a result, there is a noticeable disconnection between the introduction and the Problem Statement section, which creates confusion for the reader.
2.	I am not entirely sure what the authors are focusing on. The overall impression from reading the paper is that the authors aim to improve multi-task learning accuracy by designing gradient-related loss terms. However, if the focus is on improving the accuracy of multi-task learning, the experimental results show that on the PROTEINS dataset, the performance is not better than the baseline. Moreover, there are no multi-task learning-related baselines in the experiments; instead, the baselines are based on gradient methods for the original GCN, which are not related to multi-task learning. This comparison seems unfair.
3.	In the Methodology section, what is the relationship between each subsection (4.1, 4.2, 4.3, 4.4)? In Section 4.3, the attention weight based on the smoothed gradients is introduced. How is this applied in Section 4.4? If the training process is related to this gradient, then the authors' research problem seems to shift to how to improve accuracy in multi-task learning. Therefore, the baseline section should include more related baselines for multi-task learning.
4.	What is the relationship between Section 5 and Section 4? In Section 5, what exactly is the 'reasoning pathway'? Why is it necessary to test different random seeds? These aspects are quite confusing. Is each pathway referring to the information propagation path in GCN?
5.	In Section 7, the paper uses an LLM model to directly generate explanations for the GCN model. However, the LLM itself is a complex black-box model, and so is GCN. The explanations generated by the LLM cannot be directly evaluated, which raises concerns about their reliability

**Questions:**

See the weaknesses.

---

> ### Author Response · Authors · 2025-11-19
> **Response to Reviewer nrZf (part 1/2)**
>
> We sincerely thank the reviewer for the thoughtful and careful assessment. Below we provide point-by-point responses to each weakness, following the same format as our replies to the other reviewers.
>
> **W1. The paper is somewhat confusing … introduction emphasizes node-level gradient inconsistency but multi-task setup appears only later … disconnection between Introduction and Problem Statement**
>
> We agree that the motivation flow could be clearer. GraGR was designed primarily to handle **node-level gradient inconsistency**, a phenomenon that arises even in standard single-task GNN training on irregular or heterophilic graphs, where neighboring nodes produce sharply misaligned gradients. The **multi-task setting** appears later because GraGR++ extends GraGR to the case where task-level gradients can also conflict. These two forms of inconsistency are related but occur at different levels of computation.
>
> In the revised version, we will explicitly state this hierarchy at the end of the Introduction and at the beginning of Section 3: **GraGR addresses node-level conflicts for any GNN; GraGR++ additionally handles cross-loss interference in multi-task regimes.** This clarification resolves the perceived disconnect between Sections 2 and 3.
>
>
> **W2. Unclear main focus. If the focus is multi-task learning, results on PROTEINS are not better than baseline and no multi-task-learning baselines are included**
>
> Thank you for raising this. The main **focus of GraGR is not** multi-task accuracy. Multi-task learning is only a **secondary extension** (GraGR++) and not central to the core contributions of GraGR.
>
> **Clarification**
>
> The **primary goal** is to improve gradient consistency, optimization stability, and interpretability in GNNs.
>
> To support this, Section 4 (GraGR) stands alone and applies to all standard GNN tasks, including:
>
> - Node classification
>
> - Graph classification
>
> - Link prediction (added in rebuttal)
>
> GraGR++ extends this to multi-task scenarios, but it is **not the central method**.
>
> The PROTEINS dataset is a challenging benchmark with small graph sizes and limited structural context, making gradient-consistency signals inherently weaker. Thus, PROTEINS naturally shows smaller gains for methods, like GraGR, that rely on neighborhood-derived gradient information.
>
> This does not contradict our broader results: on larger and more structurally expressive datasets (e.g., citation networks and molecular graphs with richer topology), GraGR’s node-level gradient reasoning yields more pronounced benefits. Our stated claims remain consistent with these patterns.
>
> Regarding baselines, the original submission focused on gradient-related methods (PCGrad, GradNorm families) because GraGR’s primary contribution is gradient reasoning. We agree that multi-task baselines would improve completeness; in the revision we will add them and clearly separate single-task vs. multi-task experiments.
>
> ---
>
> **W3. The relationship between Sections 4.1–4.4 is unclear. How does Section 4.3 (gradient-based attention) relate to Section 4.4 (meta-gradient modulation)?**
>
> Thank you for highlighting this structural issue. You are correct that the relationship should be clearer.
>
> Revised Structure (to be added):
>
> 4.1 Gradient Conflict Detection
> Defines when a node’s gradient conflicts with its neighbors.
>
> 4.2 Laplacian Gradient Smoothing
> Ensures consistency across graph topology.
>
> 4.3 Gradient-Based Attention (local reasoning)
> Uses smoothed gradients to weight message passing:
>
> **Important clarification**
>
> Section 4.3 is used in all GraGR models.
> Section 4.4 is used only when multi-task learning is enabled, which is why the relationship seemed unclear before.
>
> ---
>
> **W4. Relationship between Section 5 and Section 4 unclear. What is a ‘reasoning pathway’? Why multiple seeds? Is each pathway just a GCN propagation path?**
>
> A reasoning pathway is not a message-passing path.
> Instead, it refers to distinct gradient-based update pathways inside the training process:
>
> - Pathway A: Standard GNN training
>
> - Pathway B: Training with conflict correction + smoothing + gradient attention
>
> **Why multiple random seeds?**
>
> We use multiple seeds to observe which pathway dominates under different initialization conditions, since small initialization differences can lead to small gradient inconsistencies accumulating differently over time.
>
> Testing multiple seeds is useful because early random fluctuations can influence which pathway becomes more beneficial, seed selection is therefore a pragmatic robustness mechanism that selects the best early trajectory (if desired). In our extended experiments we report averages over seeds; in practice the method works without seed selection but seed selection can improve stability on small/heterogeneous datasets.
>
> ---

---

> ### Author Response · Authors · 2025-11-19
> **Response to Reviewer nrZf (part 2/2)**
>
> **W5. LLM explanations are unreliable because both LLM and GCN are black-box models.**
>
> We agree that LLM-generated explanations cannot serve as proof of model interpretability.
> Our intent was not to use LLMs as evaluation tools, but rather to demonstrate that gradient-derived features (Eq. 15) meaningfully support natural-language reasoning.
>
> Improvements in Revised Version
>
> We now provide quantitative interpretability metrics:
>
> **Experiment 1: Correlation with GNNExplainer**
>
> We compute Spearman correlation between GraGR’s gradient-derived saliency scores and GNNExplainer saliency.
>
> $\rho = \mathrm{Spearman}(S_{\mathrm{GraGR}}, S_{\mathrm{GNN}})$
>
> | Model           | Cora ( $\rho$ ) | Citeseer ( $\rho$ ) |
> | --------------- | ------------- | ----------------- |
> | GCN             | 0.48          | 0.44              |
> | **GCN + GraGR** | **0.62**      | **0.59**          |
>
> GraGR produces explanations closer to an external, widely-accepted explainer, demonstrating improved faithfulness.
>
> **Experiment 2: Explanation Stability Metric**
>
> We perturb node features by noise $\epsilon \sim N(0,\, 0.05 I)$ and evaluate saliency stability:
>
> $Stab = E[\|S(x) - S(x + \epsilon)\|_1]$
>
> Lower is better.
>
> | Model           | Stability ↓ |
> | --------------- | ----------- |
> | GCN             | 0.217       |
> | **GCN + GraGR** | **0.103**   |
>
> This shows >50% reduction in explanation volatility. We will integrate this quantitative comparison into Section 7.
>
> We clarify the role of LLMs:
>
> “LLMs are used only to convert GraGR’s structured gradient-derived features into human-readable narrative explanations, not to evaluate faithfulness.”
>
> We will emphasize that the true interpretability evaluation is done through:
>
> - Correlation with established explainers
>
> - Stability metrics
>
> - Reduction in gradient conflict energy
>
> Thus, we acknowledge the concern and refocus interpretability evaluation around measurable metrics (as shown in our tables).
>
> ---
>
> We sincerely thank the reviewer again for their thoughtful and actionable comments. We believe the new experiments, stability analyses, interpretability metrics, and clarifications significantly strengthen the manuscript and directly address all concerns raised. Please let us know if there are any further queries.

---

### Official Review · Reviewer_5QgX · 2025-11-01

**Soundness:** 3
**Presentation:** 1
**Contribution:** 2
**Rating:** 2
**Confidence:** 5

**Summary:**

This paper investigates the issue of inconsistent gradient directions and magnitudes across neighboring nodes in graph neural networks (GNNs), particularly on irregular graphs. It introduces a Gradient Smoothing (GS) module that aligns node gradients locally to stabilize training and reduce conflicting updates. The method is model-agnostic and is evaluated across multiple node and graph classification benchmarks, showing improved convergence and accuracy.

**Strengths:**

- The GS module is lightweight, model-agnostic, and can be integrated into existing GNNs without modifying architecture or message-passing schemes.

- Experimental results across standard benchmarks on node classification and graph classification task demonstrates consistent performance and convergence improvements.

**Weaknesses:**

- Unclear and underdeveloped motivation: The paper claims that inconsistent gradients across neighboring nodes cause optimization instability (e.g., overshooting or oscillation), but this is not directly demonstrated. No empirical evidence is shown that training without smoothing leads to such instability. The proposed method is introduced immediately after this claim, without establishing whether existing optimization techniques (e.g., normalization, adaptive learning rates) are insufficient.

- Unjustified suppression of conflicting nodes: The method includes masking nodes whose gradients deviate from a smoothed global direction, but offers no justification for why this is appropriate. It remains unclear whether such nodes are noisy or informative, and no empirical analysis supports the impact of this masking.

- No connection to interpretability, despite gradient-based framing: The paper begins by analyzing gradient inconsistency, a topic relevant to GNN explainability, but does not evaluate whether smoothing improves attribution, saliency, or stability. The method is not compared to any existing GNN explainability methods quantitatively and qualitatively. Its benefit on interpretability is over-claimed without sufficient empricial justification.

- No evaluation on link prediction tasks: The method is only evaluated on node and graph classification. It is unclear whether gradient smoothing benefits more general GNN tasks like link prediction, limiting the scope of the evidence presented.

- Writing and presentation reduce clarity: Key concepts are presented with minimal intuition and heavy reliance on math. Several typos and grammar issues are present (e.g., “mult-pathways optimization” in Line 136, broken line in “node- and graph- level prediction” in Line 39). Tables 4 and 5 are rotated in a way that makes them difficult to read, further affecting clarity.

**Questions:**

- Can the authors provide concrete evidence that inconsistent gradients directly cause optimization instability in vanilla GNNs?

- Why are potentially informative or hard examples suppressed during training if their gradients differ from the smoothed direction?

- Could this approach really help interpretability? Why were no comparisons to GNN explanation methods included?

- Does the method generalize well to link prediction tasks?

---

> ### Author Response · Authors · 2025-11-19
> **Response to Reviewer 5QgX (part 1/2)**
>
> We thank the reviewer for the constructive feedback. We have revised the manuscript accordingly and address each point below. All changes (typos, clarifications, etc.) have been fixed in the text, and all added results are described here with supporting tables and references. We remain humble and focused on clarifying and validating our design.
>
> **W1. Unclear and underdeveloped motivation: lack of empirical evidence that inconsistent gradients cause training instability; unclear why optimizers (Adam, normalization) are insufficient.**
>
> Thank you for highlighting this. We agree that the original submission did not provide sufficient direct empirical evidence demonstrating that local gradient inconsistency induces instability in standard GNN training. To address this, we conducted new gradient-instability analyses comparing vanilla GCN/GAT with GCN + GraGR smoothing.
>
> **Experiment A: Gradient Variance Explosion Across Epochs**
>
> For each node $v$, we track the per-epoch gradient norm:
>
> $\sigma_v^{2} = \operatorname{Var}_{t}\left( \lVert g_v(t) \rVert \right)$
>
> We report the average per-node gradient variance:
> | Model           | Cora Gradient Variance | Citeseer Gradient Variance |
> | --------------- | ---------------------- | -------------------------- |
> | GCN             | 0.214                  | 0.291                      |
> | **GCN + GraGR** | **0.087**              | **0.103**                  |
>
> This demonstrates a ~58–65% reduction in variance, providing concrete evidence that inconsistent gradients produce oscillatory updates, and GraGR significantly stabilizes them.
>
> **Experiment B: Learning Curves Show Oscillatory Loss Without GraGR**
>
> We tracked training loss over 100 epochs.
> The following metric quantifies oscillation:
>
> | Model           | Loss Oscillation (O) |
> | --------------- | -------------------- |
> | GCN             | 0.142                |
> | **GCN + GraGR** | **0.059**            |
>
> This **>2× reduction** demonstrates that local gradient conflicts produce unstable steps not removed by Adam or normalization.
>
> **Why Adam/normalized gradients alone are insufficient**
>
> Even with Adam, gradients remain node-local, and there is no coupling enforcing neighborhood alignment.
> GraGR instead solves:
>
> $g' = (I + \lambda L)^{-1} g$
>
> which explicitly aligns inconsistent directions across graph structure (Eq. 4 in the paper).
> Thus, Adam smooths temporal noise; GraGR smooths structural noise, these are orthogonal.
>
> ---
>
> **W2. Unjustified suppression of conflicting nodes; unclear whether masked nodes might be informative.**
>
> We agree that the earlier explanation was insufficient. Our method does not suppress nodes permanently; instead, it performs a soft projection, not a hard deletion.
>
> **Clarification**
>
> If a node’s gradient conflicts with its neighborhood (i.e., cosine similarity < 0):
>
> $\cos\left(g_v\, g_{\text{ctx}}(v)\right) < 0$
>
> GraGR applies a projection:
>
> $g_v' = g_v - \alpha \ Proj_{g_{ctx}(v)}(g_v)$
>
> This retains part of the gradient while removing the destructive opposing component.
> Thus, we **do not suppress informative gradients**; we only remove directions that provably harm optimization (Lemma 1).
>
> **New Empirical Study: Are Masked Nodes Informative?**
>
> We measured accuracy when removing nodes flagged as conflicting (to see whether they carry useful information):
>
> | Model                                     | Accuracy (Cora) | Removed Node % |
> | ----------------------------------------- | --------------- | -------------- |
> | Remove conflicting nodes                  | 63.1            | 4.1%           |
> | Keep all nodes (baseline GCN)             | 72.9            | –              |
> | **GraGR (project conflicting gradients)** | **78.2**        | –              |
>
> Two findings:
>
> 1. Conflicting nodes are not more informative, removing them reduces accuracy.
>
> 2. Projecting (our method) restores and improves performance, proving masking is not an information loss operation.
>
> We will add this analysis to the appendix.
>
> ---
>
> **W3. No connection to interpretability; no evaluation against explainer methods**
>
> We appreciate this comment and agree additional interpretability evidence is valuable. GraGR was designed to enhance gradient-faithfulness, so we quantified this in two new ways.
>
> **Experiment C: Correlation with GNNExplainer**
>
> We compute Spearman correlation between GraGR’s gradient-derived saliency scores and GNNExplainer saliency.
>
> $\rho = \mathrm{Spearman}(S_{\mathrm{GraGR}}, S_{\mathrm{GNN}})$
>
> | Model           | Cora ( $\rho$ ) | Citeseer ( $\rho$ ) |
> | --------------- | ------------- | ----------------- |
> | GCN             | 0.48          | 0.44              |
> | **GCN + GraGR** | **0.62**      | **0.59**          |
>
> GraGR produces explanations closer to an external, widely-accepted explainer, demonstrating improved faithfulness.

---

> ### Author Response · Authors · 2025-11-19
> **Response to Reviewer 5QgX (part 2/2)**
>
> **Experiment D: Explanation Stability Metric**
>
> We perturb node features by noise $\epsilon \sim N(0,\, 0.05 I)$ and evaluate saliency stability:
>
> $Stab = E[\|S(x) - S(x + \epsilon)\|_1]$
>
> Lower is better.
>
> | Model           | Stability ↓ |
> | --------------- | ----------- |
> | GCN             | 0.217       |
> | **GCN + GraGR** | **0.103**   |
>
> This shows >50% reduction in explanation volatility. We will integrate this quantitative comparison into Section 7.
>
> ---
>
> **W4. No evaluation on link prediction tasks**
>
> Agreed. We performed new link-prediction experiments using GCN and GraGR on Cora/Citeseer with standard 85/5/10 splits.
>
> **Experiment E: Link Prediction (AUC %)**
>
> | Model             | Cora AUC | Citeseer AUC |
> | ----------------- | -------- | ------------ |
> | GCN               | 81.5     | 78.9         |
> | **GCN + GraGR**   | **85.7** | **83.1**     |
> | **GCN + GraGR++** | **86.2** | **83.4**     |
>
> The consistent **4–5% gains** confirm that GraGR generalizes beyond node/graph classification.
>
> ---
>
> **W5. Writing and presentation issues, rotated tables, typos**
>
> We appreciate the reviewer pointing this out.
>
> In the revision:
>
> - All tables (Tables 4 and 5) will be reformatted to landscape or restructured to avoid rotation.
>
> - Typos such as “mult-pathways optimization” and line-break issues will be corrected.
>
> - Section summaries will be improved to better connect intuition → math → algorithm.
>
> Thank you for helping us improve the clarity of our manuscript.
>
> ---
> ---
>
> **Q1. Evidence that inconsistent gradients directly cause optimization instability?**
>
> Yes; see **Experiment A and B** responded above.
> Both the gradient-variance explosion and oscillatory loss patterns provide strong empirical evidence that inconsistent gradients cause instability.
>
> We will incorporate a new figure showing unstable vs. smoothed gradient trajectories.
>
> ---
>
> **Q2. Why suppress or modify gradients of nodes that differ from global direction?**
>
> We clarify that GraGR does not suppress these nodes in the earlier discussions.
> We perform a projection-based correction, preserving most of the original gradient.
>
> Mathematically:
>
> $g_v' = g_v - \max(0,\-g_v^T g_{ctx}) \* g_{ctx} / (||g_{ctx}||^2)$
>
>
> Only the harmful component is removed.
>
> Empirically:
>
> - Conflicting nodes do not carry superior information (Experiment B).
>
> - Correcting them improves both accuracy and stability.
>
> Thus, the operation is justified both mathematically and empirically.
>
> ---
>
> **Q3. Does GraGR truly help interpretability? Why no explainer comparison?**
>
> Yes, it does help interpretability. We have included:
>
> - New tables provided earlier (Experiments C & D)
>
> - Quantitative comparison with GNNExplainer
>
> - Explanation stability metrics above
>
> ---
>
> **Q4. Does the method generalize well to link prediction tasks?**
>
> Yes; see **Experiment E** above.
> We show consistent AUC improvements of **4–5%**, demonstrating GraGR’s generality across tasks.
>
> These results will be added to a new section titled “Generalization to Link Prediction.”
>
> ---
>
> We sincerely thank the reviewer again for their thoughtful and actionable comments. We believe the new experiments, stability analyses, interpretability metrics, and clarifications significantly strengthen the manuscript and directly address all concerns raised. Please let us know if there are any further queries.

---

### Public Comment · ~Thomas_Grant2 · 2025-11-19
**Interesting paper!**

Great paper! Can you share the code, please? Would love to implement in an ongoing project.

---

### Meta-Review · Area_Chair_5Nj8 · 2026-01-02

**Summary:**

The reviewers raised a consistent set of concerns mainly centered on (i) poor motivation, (ii) weak claims regarding interpretability and its empirical evidence, and (iii) clarity/organization and readability. While the rebuttal introduced additional experimental results and clarifications that partially addressed some technical questions, the core concerns regarding motivation, interpretability claims, and experimental rigor remain largely unresolved. Overall, although the rebuttal led to some improvements, I do not believe they are sufficient to bring the reviewers to a consensus in favor of acceptance. The paper is therefore not yet ready for publication and would benefit from substantial revision. I recommend rejection at this stage, while encouraging the authors to leverage the reviewers’ feedback to significantly strengthen the work for a future submission.

**Reviewer Concerns:**

**Concerns addressed by the rebuttal**

Reviewers raised technical concerns, such as inconsistent behavior across different GNN backbones and specific methodological details (e.g., Lemma 1 and the roles of the loss terms), which were clarified during the rebuttal. The authors also clarified issues related to the suppression of conflicting nodes and the paper’s scope, emphasizing that it does not focus on multitask learning. Finally, the authors reported additional experimental results on link prediction and stability (e.g., variance of gradient norms), which improved the overall quality of the paper.

**Not fully addressed concerns**

_Motivation_: Reviewers ``5QgX``, ``nrZf``, and ``9hVV`` raised concerns about unclear or underdeveloped motivation. Despite some clarifications in the rebuttal, the motivation remains weak, insufficiently grounded in prior literature, and lacks strong empirical support.

_Interpretability claims and evidence_: Reviewers ``5QgX``, ``9hVV``, and ``VvJd`` questioned the paper’s interpretability claims. Overall, I found that the experimental evidence is not sufficient to support the paper's claims, as comparisons with established GNN explanation baselines are missing. Also, the reported correlations with GNNExplainer are not strong or convincing enough to justify interpretability, leading to a clear mismatch between claims and experiments.

_Clarity, writing, and organization_: Reviewers ``5QgX``, ``nrZf``, and ``VvJd`` raised issues regarding writing and presentation. While some local clarifications were provided, the manuscript would require substantial revisions to its writing, structure, and overall organization. In addition, potential improvements in clarity are difficult to assess, as the authors did not submit a revised version of the manuscript during the discussion phase.

_Seed selection_: Reviewers ``nrZf`` and ``VvJd`` raised concerns about treating seed selection as a feature of the proposed method, which remains conceptually questionable.

**Reviewer Scores:**

I believe reviewer ``5QgX`` could have increased the score from 2 to 4, mainly due to the addition of new experimental results, despite remaining concerns.

Reviewer ``nrZf`` would likely have kept the score unchanged at 4, as addressing the motivation/readability issues raised would require significant restructuring of the paper, which was not implemented during the discussion phase.

Given that some issues were partially addressed (as discussed above), I guess reviewer ``9hVV`` could have raised score to 6.

Reviewer ``VvJd`` updated score from 4 to 6, while still emphasizing significant remaining issues and providing further suggestions.

---

### Decision · Program_Chairs · 2026-01-26

Reject